# Using macromolecular electron densities to improve the enrichment of active compounds in virtual screening

Wenzhi Ma[1,4], Wei Zhang[2,3,4], Yuan Le[1,4], Xiaoxuan Shi[1], Qingbo Xu[1], Yang Xiao[1], Yueying Dou[1], Xiaoman Wang[1], Wenbiao Zhou[1], Wei Peng [2,3], Hongbo Zhang [1✉] & Bo Huang [1✉]

The quest for effective virtual screening algorithms is hindered by the scarcity of training data, calling for innovative approaches. This study presents the use of experimental electron density (ED) data for improving active compound enrichment in virtual screening, supported by ED's ability to reflect the time-averaged behavior of ligands and solvents in the binding pocket. Experimental ED-based grid matching score (ExptGMS) was developed to score compounds by measuring the degree of matching between their binding conformations and a series of multi-resolution experimental ED grids. The efficiency of ExptGMS was validated using both in silico tests with the Directory of Useful Decoys-Enhanced dataset and wet-lab tests on Covid-19 3CLpro-inhibitors. ExptGMS improved the active compound enrichment in top-ranked molecules by approximately 20%. Furthermore, ExptGMS identified four active inhibitors of 3CLpro, with the most effective showing an $IC_{50}$ value of 1.9 μM. We also developed an online database containing experimental ED grids for over 17,000 proteins to facilitate the use of ExptGMS for academic users.

[1] Beijing StoneWise Technology Co Ltd., Haidian Street #15, Haidian District, 100080 Beijing, China. [2] State Key Laboratory of Respiratory Disease, First Affiliated Hospital of Guangzhou Medical University, 510182 Guangzhou, China. [3] Innovation Center for Pathogen Research, Guangzhou Laboratory, 510320 Guangzhou, China. [4]These authors contributed equally: Wenzhi Ma, Wei Zhang, Yuan Le. ✉email: zhanghongbo@stonewise.cn; huangbo@stonewise.cn

Over the past decade, high-throughput virtual screening has become a popular method for discovering hit compounds in the field of drug design[1–3]. When a receptor's three-dimensional (3D) structure is available, molecular docking is used to identify potential binders for the target pocket[4]. However, due to the simplifications made to achieve high computational speed, such as treating the protein as mostly rigid and handling the solvent crudely[5], docking and scoring accuracy is still suboptimal and has room for improvement. Numerous attempts have been made to address these challenges by focusing on algorithm and calculation protocol optimizations. For example, ensemble docking and induced-fit docking attempt to consider the flexibility of the pocket[6], while molecular mechanics/generalized-Born surface area method (MM/GBSA) considers the effect of solvation[7]. Moreover, various designs for scoring functions have been created by considering more ligand–protein interactions, or by training machine-learning models with structural features, and biochemical and biophysical assay results as labels[8,9]. Despite the success of these approaches in improving active compound enrichment for docking results, virtual screening still has a relatively low success rate, and more effective approaches are imperative. Since most of these approaches are designed from the perspective of algorithm and calculation protocol optimizations—and are approaching a bottleneck due to lack of training data—it is important to consider alternative perspectives by leveraging additional information that can experimentally reflect the dynamics of ligands and solvents.

Electron density (ED) maps from X-ray crystallography and Coulomb potential maps from cryogenic electron microscopy (Cryo-EM) are experimental data that provide valuable information about the dynamics of macromolecular systems, including the ligands and solvents present in the pocket[10,11]. Some studies have explored the use of these maps for intermolecular non-covalent interaction (NCI) identification[12], artificial intelligence (AI)-based molecule generation[13], and quantum mechanics parameter refinement[14]. Despite these advancements, the current virtual screening approaches rely predominantly on static structures and implicit solvent models. Thus, there is an urgent need to establish an efficient method for using these maps in docking-based virtual screening to enhance active compound enrichment.

In this study, we present a novel method, ExptGMS (Experimental ED-based Grid Matching Score), which utilizes experimental ED maps to screen docking poses for better enrichment of active compounds. A machine-learning model was built for the effective use of ExptGMS generated from multi-resolution ED maps. When tested using the Directory of Useful Decoys–Enhanced (DUD-E) dataset[15], ExptGMS displayed the ability to complement molecular docking technology by achieving an over 20% increase in active compound enrichment in the top 10, 50, and 100 ranked compounds without affecting the diversity of the screening results. Approaches like 2D and 3D molecular similarity comparisons and MM/GBSA were used as benchmarks in our study. To further confirm the real-world effectiveness of ExptGMS, we performed virtual screening for Covid-19 3CLpro inhibitors. Using a biochemical assay, we tested the protease inhibition activity of the top-ranked compounds and discovered that the combination of ExptGMS and docking score provided three times more active compounds than the use of docking score alone. Furthermore, to facilitate the use of ExptGMS by academic users, we prepared ExptGMS grids for over 17,000 proteins and developed a database that provides web-based services (https://exptgms.stonewise.cn/#/create).

## Results

**Construction of ExptGMS**. X-ray diffraction of macromolecular crystals generates an average ED over numerous crystal cells, which represents a time-average distribution of the molecules in the crystal. As shown in Fig. 1a, two ligand conformations were observed in the binding pocket. In addition, some solvent molecules with relatively intense dynamics may exhibit low intensity due to time-averaged effects, and may get overlooked during model building (Fig. 1b), resulting in incomplete modeling of the pocket contents. Given that most computational methods for virtual screening rely on static or incomplete models, the full profile of the pocket contents and their dynamic information embedded in the experimental ED maps are considered important complements to the methods currently in use.

To fully utilize the time-averaged signals in ED maps, we developed ExptGMS, which has two key components: an experimental ED-based grid and a scoring function. We used $2F_o–F_c$ ED maps with above-zero contour levels (>0 σ) for grid generation. To avoid excessive experimental noise, ED lower than zero σ were excluded. Grid points were placed in and around the pocket, and were assigned values reflecting the ED intensity at that position (Fig. 1c). A given ligand conformer is scored based on its degree of matching with the grid. In general, we developed a scoring function based on three principles: (1) rewarding ligand atoms occupying grid points with strong ED intensity; (2) penalizing ligand atoms occupying space with no grid points; (3) penalizing grid points with strong ED intensity but not occupied by any ligand atom. Details regarding the grid construction and scoring function development can be found in "Methods".

To address the bias introduced by the grid being constructed on the ED of binders and solvents observed in a limited number of experiments with a limited range of binder types, we used the concept of ED map resolution. An ED map with lower resolution contains fewer details and is more abstract than the one with higher resolution; thus, it can provide more generalized conformation matches and consequently enhance the diversity of matched molecules. For pilot testing, we chose a median resolution of 3.0 Å to create ExptGMS.

**Performance of 3.0 Å resolution ExptGMS on DUD-E dataset**. The evaluation of ExptGMS generated using experimental ED with 3.0 Å resolution was described from three perspectives: dataset, benchmark technologies, and evaluation framework.

From a total of 102 targets in the DUD-E dataset, 85 targets were selected, since the remaining had no qualified experimental ED available. For each target in the dataset, about 13,000 compounds were available, with an active-to-decoy ratio of 1:30. The binding positions of the active compounds and decoys in the corresponding pockets were obtained from a previous study which used GlideSP for docking[16].

GlideSP was selected as the docking program in our study because of its widespread use in the industry. We included two types of benchmarks for comparison with ExptGMS: pocket-based and ligand-based approaches. For the pocket-based approach, MM/GBSA was used for binding energy-focused evaluation; while pocket shape-focused evaluation was done using alpha spheres and ExptGMS shape-only (ExptGMS grid with all grid point intensities set as one). For the ligand-based approach, extended-connectivity fingerprint (ECFP) with Tanimoto index was included as the 2D similarity descriptor. In addition, Ultrafast Shape Recognition with CREDO Atom Types (USRCAT)[17]—a 3D similarity method that incorporates information on shape and pharmacophores—was used as the 3D similarity descriptor. Furthermore, three-dimensional force field fingerprint (TF3P)[18]—a newly developed 3D fingerprint for small molecules—was also included to represent the force field and deep learning-based approaches. Finally, because our goal was to test whether the addition of ExptGMS could assist docking

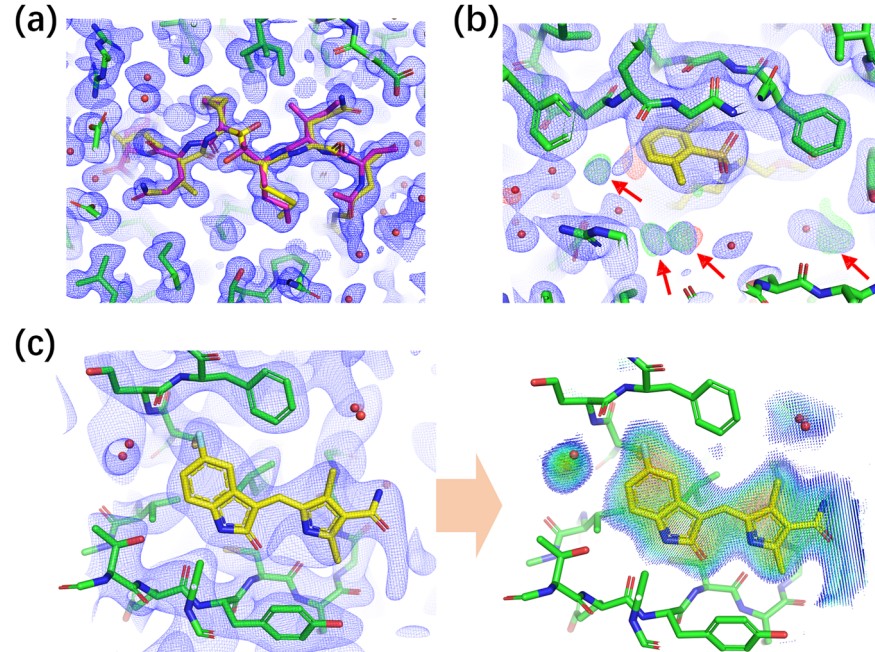

**Fig. 1 Time-averaged information embedded in experimental electron density (ED) maps. a** Two conformations of a ligand identified in an experimental ED, denoted by yellow and purple colors, respectively (PDB ID: 6KMP, resolution: 1.3 Å). **b** Additional solvent molecules indicated in ED maps (PDB ID: 3QKK, resolution: 2.3 Å). Modeling of the solvent in the pocket is incomplete due to the missing solvent molecules (indicated with red arrows).
**c** Construction of grids using experimental ED (PDB ID: 3G0E, resolution: 3.0 Å). The grids only cover the regions inside and around the pocket, excluding the region occupied by the pocket itself (i.e., protein atoms). Grid points are colored according to their ED intensities using a rainbow scheme, where red represents high and blue represents low ED intensity. For all panels, 2Fo–Fc maps are presented in blue mesh, at a contour level of 1.0 σ. **b** Fo–Fc map is presented in green (positive) and red (negative) mesh, at a contour level of 3.0 σ and −3 σ, respectively.

procedures in eliminating false positives and false negatives, we included tandemly linked GlideSP and ExptGMS scores in the test. This hybrid approach was termed GlideSP + ExptGMS and involves the process of selecting the top 10% molecules based on their ExptGMS scores, and then ranking them according to their GlideSP scores.

For the evaluation framework, two key indexes were considered: (1) enrichment of active compounds, measured by the number of active compounds in the top 10, 50, and 100 ranked molecules, and (2) diversity of the top-ranked active compounds measured using the average Tanimoto similarity of each pair of selected active compounds. We measured both enrichment and diversity because virtual screening methods should identify a variety of scaffolds in addition to a large number of active compounds.

The results were analyzed using a two-dimensional scatter plot. The highest enrichment was achieved by 2D similarity comparison, but at the cost of a significant loss in diversity (Fig. 2, Supporting Information Fig. S1 and Supplementary Data 1). Considering both enrichment and diversity, ExptGMS outperforms most of the benchmark approaches. More importantly, Glide + ExptGMS enriched more active compounds in both top 10 and top 50 than GlideSP alone, indicating that ExptGMS is complementary to GlideSP. As a pilot test using single-resolution ExptGMS, the observed complementarity is not strong, but it confirms that our research is moving in the right direction. In addition, the significant drop of performance of ExptGMS-shape-only relative to ExptGMS confirms the effectiveness of introducing ED intensity to the grid.

The complementarity of ExptGMS to GlideSP was also demonstrated in the case studies (Fig. 3). As shown in Fig. 3a, an inactive compound with a good GlideSP score was eliminated due to a poor match to the ExptGMS grid. This molecule had a

docking score of −7.3, with Rewards and HBond scores accounting for −2.9 and −1.1, respectively. As GlideSP scoring function assigns empirical terms with high weights, molecules having empirically recognizable interactions with pockets tend to score well. However, from the perspective of ExptGMS, this molecule failed to occupy a strong ED blob, resulting in a poor ExptGMS score. In addition to this false-positive elimination case, we have also listed two false-negative elimination cases. Figure 3b shows an active molecule fitted well with the ExptGMS grid, but it did not have any empirically favored interactions, and therefore had a low GlideSP score of −4.8. Furthermore, as shown in Fig. 3c, an active molecule with its carboxyl group occupying the ED originally contributed by a water molecule in the crystal structure and achieved a good ExptGMS ranking. Based solely on the low GlideSP score, this compound would have been eliminated. This case demonstrates the effectiveness of preserving the solvent ED information in the ExptGMS grid.

In summary, our pilot testing on 3.0 Å resolution ExptGMS confirmed our hypothesis that ExptGMS contains signals useful for the improvement of active compound enrichment. To further maximize the effectiveness of such signals, we considered an ExptGMS with multiple resolutions.

**Performance of multi-resolution ExptGMS on DUD-E dataset.** The ExptGMS grids display varying resolutions, much like the experimental EDs which can also vary in resolution. As shown in Fig. 4a, an ExptGMS grid with a specific resolution can be constructed using the ED map at that resolution. Furthermore, the curve in Fig. 4b illustrates that decreasing resolution results in a more uniform distribution of grid values, suggesting a higher degree of tolerance for conformational matches with ligand candidates. Such characteristics affect the recall of compounds

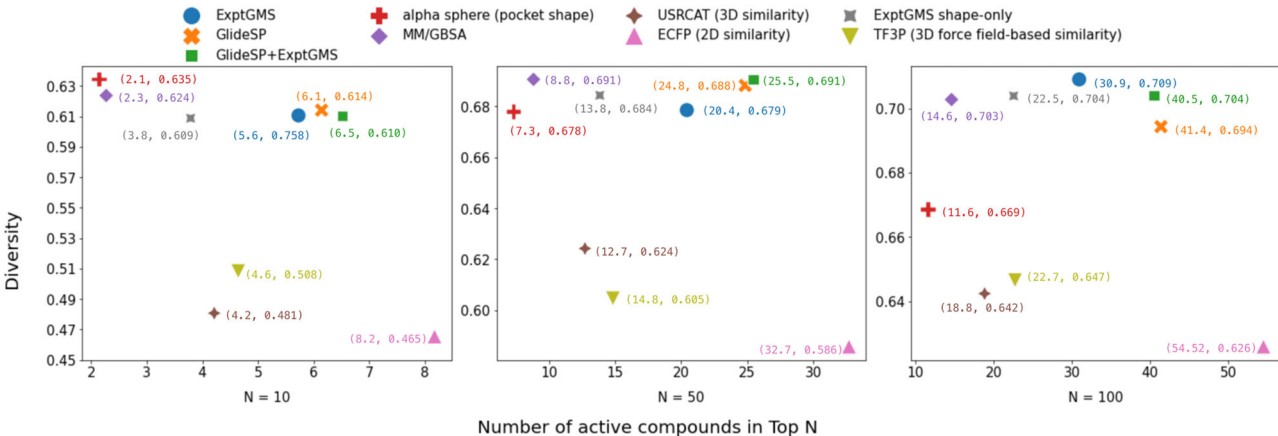

**Fig. 2 Comparison of electron density-based grid matching score (ExptGMS) with benchmark technologies using 85 targets from DUD-E dataset.** The average of pairwise Tanimoto similarities over ECFP4 fingerprints is shown for the active molecules ranked in top $N$ ($N$ = 10, 50, and 100) [diversity = 1 – (average pairwise 2D similarity among the molecules)].

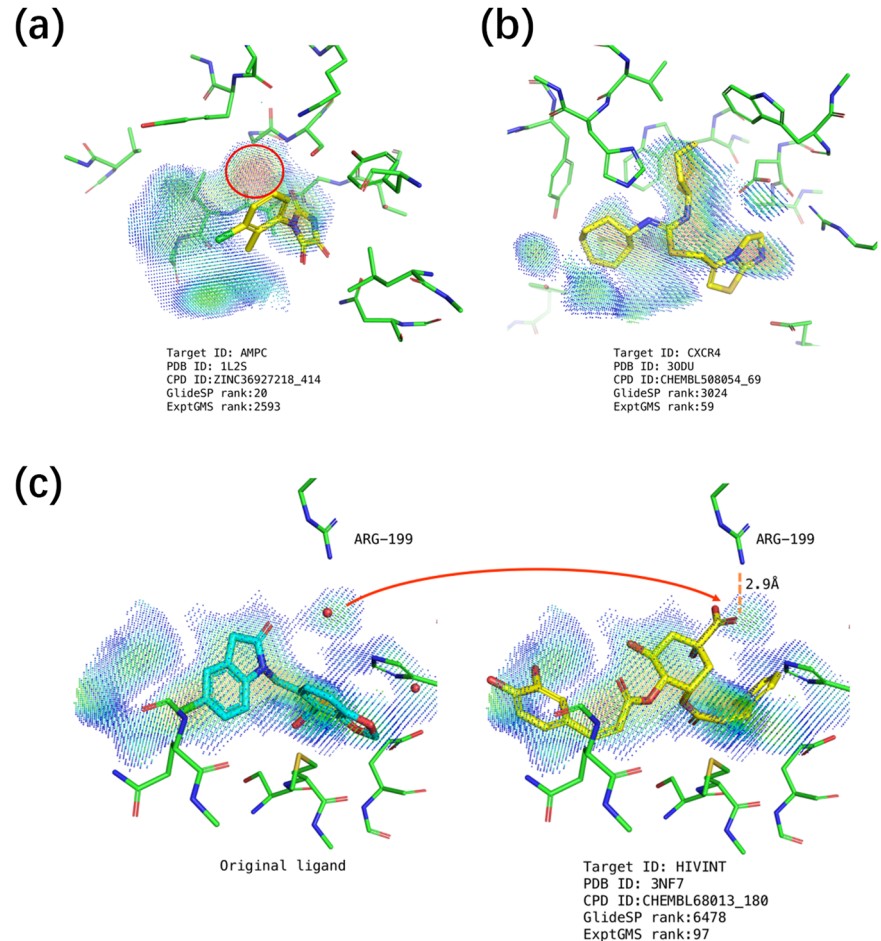

**Fig. 3 Case study of electron density-based grid matching score (ExptGMS) supporting GlideSP in eliminating false-positive and false-negative results. a** A case of false-positive elimination. The strong electron density (ED) blob missing in this molecule is indicated with a red circle. **b** A case of false-negative elimination. This active compound fits well in ExptGMS grid, but has a low GlideSP ranking. **c** Another case of false-negative elimination. The original ligand and solvent molecules observed in the crystal are shown on the left side. The carboxyl group filling the density occupied by the original water molecule is indicated by a red arrow. The ExptGMS grid points are colored according to their ED intensities using a rainbow scheme, where red represents high and blue represents low ED intensity.

that differ significantly from the reference ligand topology, and may consequently improve enrichment.

To quantify the ability of ExptGMS with varying resolutions in enriching active compounds, we extended the aforementioned pilot testing on 3.0 Å resolution ExptGMS to four additional resolutions—2.5 Å, 3.5 Å, 4.5 Å, and 5.5 Å. To enhance clarity, we listed all 85 tested targets in a circle and colored the targets using a resolution-specific color, if the active compound enrichment of

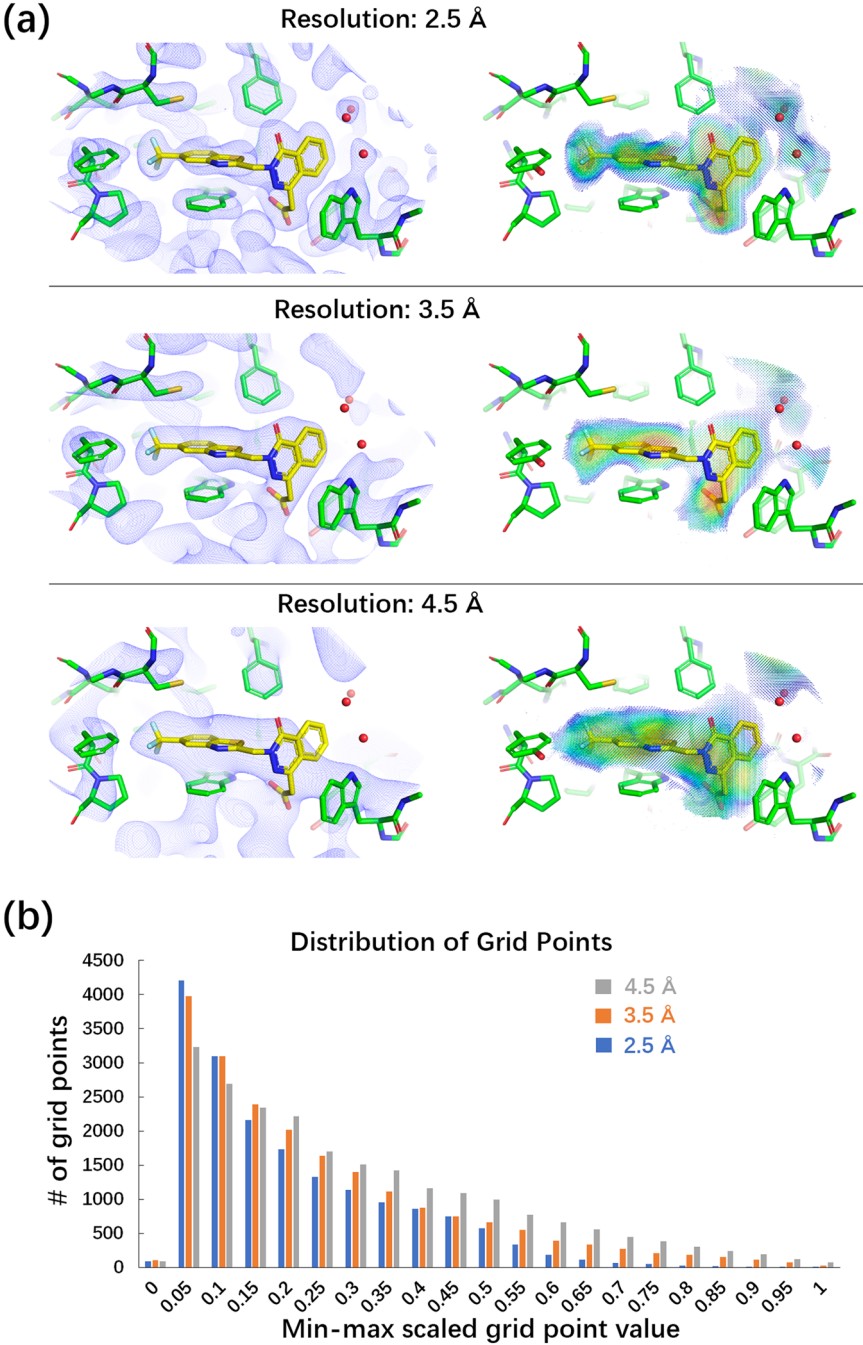

**Fig. 4 Electron density-based grid matching score (ExptGMS) grids with different resolutions. a** Multiple resolution ExptGMS grids generated using experimental electron densities (EDs) at that resolution. Experimental ED are 2Fo–Fc maps contoured at 1σ. ExptGMS grids are colored with a rainbow scheme ranging from low (blue) to high ED intensity (red). **b** Distribution of grid points by value at different resolutions. The bar chart is created with grid value normalized using a min-max scaling.

ExptGMS+GlideSP at that resolution outperformed GlideSP. Figure 5a displays the union of these colored targets across different resolutions, covering ~75% of the targets, while the intersection of these colored targets accounts for only one-third of the total. This observation suggests the potential of using multi-resolution ExptGMS to achieve superior performance.

The question arises as to why ExptGMS with different resolutions can complement each other in terms of enriching active compounds. One possible explanation is that ExptGMS with different resolutions intend to score ligands from different perspectives. Low-resolution grids focusing on scaffold-level information, whereas high-resolution grids focusing on R group

of atomic-level information. This distinction arises due to the intrinsic characteristic of X-ray or electron diffraction-based density, where decreasing the resolution results in a more uniform intensity distribution with fewer details expressed in the density map. To illustrate this point, we present a case involving PDB ID 2HV5. Here, an active compound exhibits a similar binding mode and scaffold to the co-crystallized ligand of the protein (Fig. 5b). This active ligand (yellow) can be ranked in top 100 by using ExptGMS with 3.5 Å but not with 2.5 Å. To highlight the difference of ExptGMS grids at these two resolutions, we selected grid points with strong intensities (i.e., over 3 σ) and showed them side by side in Fig. 5c. The 2.5 Å grid

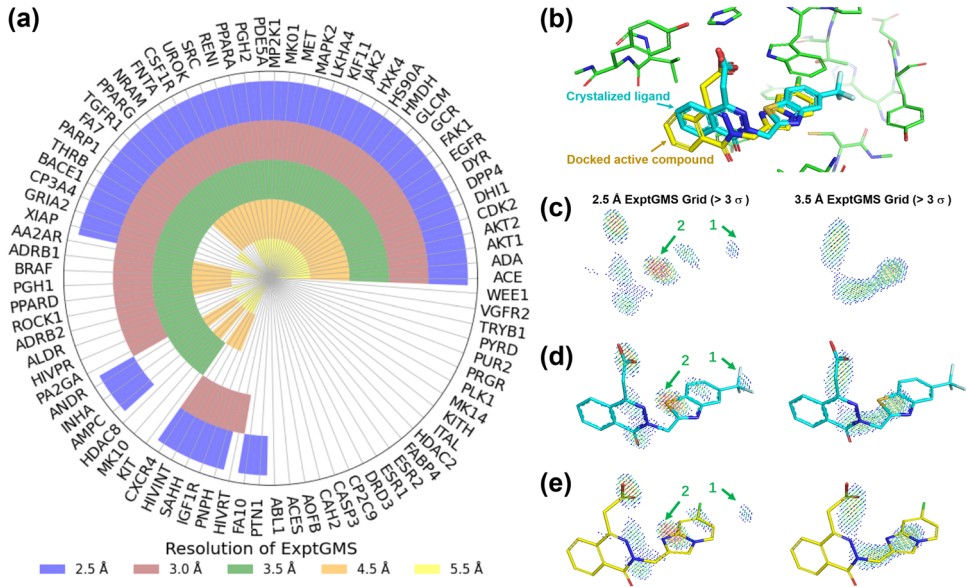

**Fig. 5 Performance of electron density-based grid matching score (ExptGMS) with varying resolutions on 85 targets from the Directory of Useful Decoys–Enhanced (DUD-E) dataset. a** Performance comparison of ExptGMS at different resolutions. A target was labeled with a resolution-specific color if ExptGMS+GlideSP demonstrates more active compounds than GlideSP score alone, among any of the top 10, 50, or 100 ranked compounds for that target, at that particular resolution. **b** The binding mode of co-crystallized ligand and docked active compound (CHEMBL344526_63) in the pocket of PDB ID 2HV5. The co-crystallized ligand, active compound, and pocket atoms are colored with cyan, yellow, and green, respectively. **c** ExptGMS grids of 2HV5 pocket at 2.5 Å and 3.5 Å. Only the grid points with ED intensity over 3.0 σ are shown. ExptGMS grids are colored with a rainbow scheme ranging from low (blue) to high ED intensity (red). The outlying blob (#1) and strong-grid-points-concentrated blob (#2) are indicated with green arrows to show the fragmentation and disparate distribution of 2.5 Å grids, respectively. **d** The match of co-crystallized ligand with ExptGMS grids at 2.5 Å and 3.5 Å. The molecule in cyan matches with 2.5 Å grid better than 3.5 Å. **e** The match of active compound (CHEMBL344526_63) with ExptGMS grids at 2.5 Å and 3.5 Å. The yellow molecule exhibits a better match with the 3.5 Å grid: it attains a top 100 ranking in ExptGMS at 3.5 Å, whereas it does not achieve a similar ranking at 2.5 Å. The penalty associated with the presence of blob #1 in the 2.5 Å grid is eliminated in the 3.5 Å grid. In addition, the expansion of blob #2 in the 2.5 Å grid across a larger area in 3.5 Å grid makes 3.5 Å grid align more favorably with the compound's scaffold.

appears more fragmented, containing numerous blobs with high intensity (red grid points) than 3.5 Å grid. When scoring the original co-crystallized ligand (cyan), the fragmented blobs in the 2.5 Å grid exhibit a higher degree of matching with the ligand than the 3.5 Å grid (Fig. 5d). However, when scoring the active compound sharing similar scaffold but with different substitution groups, the 3.5 Å grid shows better match than the 2.5 Å grid. Fig. 5e illustrates that the penalty introduced by the fragmented blob (#1) in the 2.5 Å grid is waived in the 3.5 Å grid, and the strong blob (#2) in the 2.5 Å grid spreads across a wider region, fitting more accurately with the scaffold profile of the compound.

**Multi-resolution ExptGMS-powered machine-learning model.** To further demonstrate the value of multi-resolution ExptGMS, we developed a straightforward machine-learning model using Gradient Boosting Decision Tree (GBDT) for signal integration. We did not select a more complicated model because we focused on testing the value of the data. Our GBDT model is a classification model that was trained and tested using 85 targets from the DUD-E dataset. Specifically, the training set contained 73 targets, and the test set contained 12 targets. To prevent information leakage, the division of the training and test sets (Supporting Information Table S1) was split in a way that no targets in the test set had homology with a sequence identity greater than 30% in the training set. The activities reported in the DUD-E dataset were used as labels.

The GBDT model was trained in different versions, in which the selected features provided by benchmark technologies were used. As shown in Table 1 (details in Supplementary Data 2 and 3), the combination of GlideSP and ExptGMS exhibited the highest enrichment and good diversity of active compounds. The

addition of multi-resolution ExptGMS improved the enrichment of active compounds in top 10 and top 50 by more than 20%, compared to the use of GlideSP alone.

The confidence intervals for the active compounds in the top $N$ were obtained using the bootstrapping method. Specifically, samples were randomly selected with replacement from the test dataset until the selected sample size matched the size of the test dataset. Considering all the selected samples, the average numbers of active compounds in the top 10, 50, and 100 results were calculated, respectively. By repeating this process 200 times, a distribution of results was generated. From this distribution, the mean value and percentile confidence interval were computed.

In addition to enhancing the enrichment of active compounds within the top $N$ ranked molecules, we sought to assess the impact of ExptGMS on the classification of active and decoy compounds. For evaluation, we utilized the area under the receiver operating characteristic curve (AUROC). The GBDT model incorporating both GlideSP score and ExptGMS features, demonstrated a higher AUROC compared to the model that solely utilized GlideSP score as a feature (Table 1), reflecting the classifier's improvement with the inclusion of ExptGMS. Nonetheless, it is important to acknowledge that this improvement is mild, and the absolute AUROC value remains relatively low, indicating the need to incorporate ExptGMS features into more sophisticated models in future research.

**Application of ExptGMS in virtual screening of Covid-19 3CLpro inhibitors.** To further validate the efficiency of ExptGMS in the real world, we applied this method for the virtual screening of 3CLpro inhibitors. Using the pocket structure extracted from the crystal structure of SARS-CoV-2 3CL protease (PDB ID

**Table 1 Performance of GBDT models trained using different features on the DUD-E test set ($N = 12$).**

| Features used in the model | Average number of active compound in top $N$, with 90% confidence interval | | | Diversity[*b] | AUROC[*d] |
|---|---|---|---|---|---|
| | $N = 10$ | $N = 50$ | $N = 100$ | | |
| GlideSP and multi-resolution ExptGMS[*a] | 5.4 [5.0, 5.8] | 21.6 [20.4, 22.9] | 33.2 [30.9, 35.1] | 0.64 [0.60, 0.67] | 0.66 |
| GlideSP and TF3P | 5.4 [4.9, 5.8] | 19.5 [18.1, 20.8] | 27.7 [26.0, 29.6] | 0.62 [0.60, 0.64] | 0.64 |
| GlideSP and MM/GBSA | 4.5 [4.0, 5.2] | 18.5 [17.0, 19.9] | 30.1 [28.1, 32.2] | 0.66 [0.62, 0.68] | 0.65 |
| GlideSP and USRCAT | 4.5 [3.9, 5.1] | 18.1 [16.9, 19.3] | 30.4 [28.0, 32.3] | 0.63 [0.59, 0.66] | 0.63 |
| GlideSP and Alpha sphere | 5.1 [4.6,5.6] | 18.4 [17.0, 19.8] | 29.2 [27.2, 31.0] | 0.64 [0.61, 0.68] | 0.63 |
| GlideSP[*c] | 4.3 [3.8, 4.8] | 17.3 [16.1, 18.6] | 29.9 [27.9, 31.9] | 0.66 [0.62, 0.68] | 0.62 |

[*a]Multiresolution indicates ExptGMS at resolutions of 2.5, 3.0, 3.5, 4.5, and 5.5 Å.
[*b]Diversity of active compounds among the top 100 ranked compounds.
[*c]Ranking was performed directly using the GlideSP score without using a trained model.
[*d]AUROC was calculated by using GBDT predicted probability of being an active compound.

7VU6), GlideSP-based molecular docking was performed against an 8-million-compound library compiled by consolidating commercially available compounds. Subsequently, the 3 Å resolution ExptGMS score was calculated for the conformations obtained from the molecular docking. 24 molecules were selected by intersecting the top 500 compounds ranked by ExptGMS score with the top 1000 compounds ranked by docking score. These 24 compounds were evaluated using wet-lab tests to determine their inhibitory rates and $IC_{50}$ values. The top 24 compounds, ranked solely by docking scores, were also tested to serve as controls. It is important to mention that no visual inspection or manual selection was involved in the selection of the aforementioned 48 compounds.

The structures, binding modes, and $IC_{50}$ values of the tested compounds are presented in Fig. 6 and Supporting Information (Supplementary Tables S2 and S3 and Supplementary Figs. S2 and S3). Among the 24 molecules selected using ExptGMS and GlideSP, nine molecules exhibited an inhibition rate greater than 50%, and four molecules exhibited $IC_{50}$ values of less than 25 μM, with the best one hitting 1.9 μM. In contrast, only three molecules exhibited inhibition rate greater than 50%, in the top 24 ranked molecules obtained using GlideSP alone, and only one of them exhibited $IC_{50}$ around 10 μM.

In conclusion, ExptGMS significantly enhanced the enrichment of active compounds in our Covid-19 3CLpro-inhibitor screening study.

**Construction of ExptGMS database and online service**. Despite the value of multi-resolution ExptGMS demonstrated in the above study, the construction of ExptGMS grids is not straightforward for end users. To facilitate the use of our approach by academic users, we processed multi-resolution ExptGMS grids for over 17,000 proteins and created a web-based server that can be accessed at this link (https://exptgms.stonewise.cn/#/create). Prior to working, users are required to select grids by specifying the PDB code from a drop-down list, upload the ligand poses, and upload the pocket structure which will be used to align the ligands with the grids. Typically, our service takes about one hour to complete the ExptGMS scoring for 100,000 compounds. ExptGMS grids at 2.5, 3.0, 3.5, 4.5, and 5.5 Å resolutions will be used to score the docking conformations. The ExptGMS scores at each resolution are written in the output SDF file. If a "docking_score" attribute is available in the uploaded SDF file, it will be combined with multi-resolution ExptGMS scores and submitted to our GBDT model, and the predicted probability of being an active compound will be added to the output SDF file.

**Discussion**
Our study on ExptGMS demonstrates that the use of experimental ED can improve the enrichment of active compounds in molecular docking-based virtual screening. In this section, we discuss two topics: (1) how to further leverage multiple-crystal information, if available; and (2) what limitations ExptGMS currently has, and how they can be overcome in the future.

Given that most of the popular targets have more than one available crystal structure, it is necessary to discuss whether ExptGMS can benefit from using multi-crystal information. An intuitive approach involves creating an ExptGMS grid for each crystal structure, followed by an averaging procedure. We tried such approach on four crystal structures of RAC-alpha serine/threonine-protein kinase (AKT1) and tested the performance of the multi-crystal averaged ExptGMS using active compounds and decoys of this target in DUD-E (Fig. 7). As shown in Table 2, the multiple-crystal-averaged ExptGMS significantly outperformed the single-crystal ExptGMS. Such an analysis provides a good start for future scope of investigating the optimal strategy for using ExptGMS to improve ensemble docking.

ExptGMS has three limitations. First, the current use of ExptGMS relies on the binding conformation achieved by the docking programs, which limits the value of ExptGMS if the binding pose is incorrect. An example is shown in Fig. 8, where an incorrect binding pose can be intuitively corrected by aligning the molecule to the ExptGMS grid. Real-space refinement technology[19] used in crystallography can serve as a good starting point for the development of ExptGMS-based high-speed binding-pose-search engines. Second, the construction of ExptGMS depends on the availability of experimental ED. However, these conditions cannot always be satisfied. For example, it is common to use apo-protein structures processed by molecular simulation for virtual screening, especially for studies in which allosteric pockets are considered. In this scenario, an experimental ED is not available. One plausible approach to address this challenge is to conduct co-solvent molecular dynamics[20] to find a fragment-sized binder in the potential pocket and use computational ED to construct the ExptGMS grid. Using an AI generative model[13] to create a filler ED in the pocket is also an alternative approach. Third, the current version of ExptGMS lacks information to support the estimation of the interactions between ligands and pockets. This explains why the combination of GlideSP and ExptGMS performs better than ExptGMS alone; GlideSP complements this interaction estimation. A clue to address this challenge can be found in a previously reported study on identifying NCIs between ligands and pockets by studying the saddle points of the ED[12]. Incorporating NCI-related saddle points into ExptGMS grids and assigning them appropriate weights could be a potential solution. An alternative solution could be the introduction of electrostatic surface potential (ESP)-matching score into ExptGMS grids. A previous study[21] that has discussed this topic is a good starting point for future research.

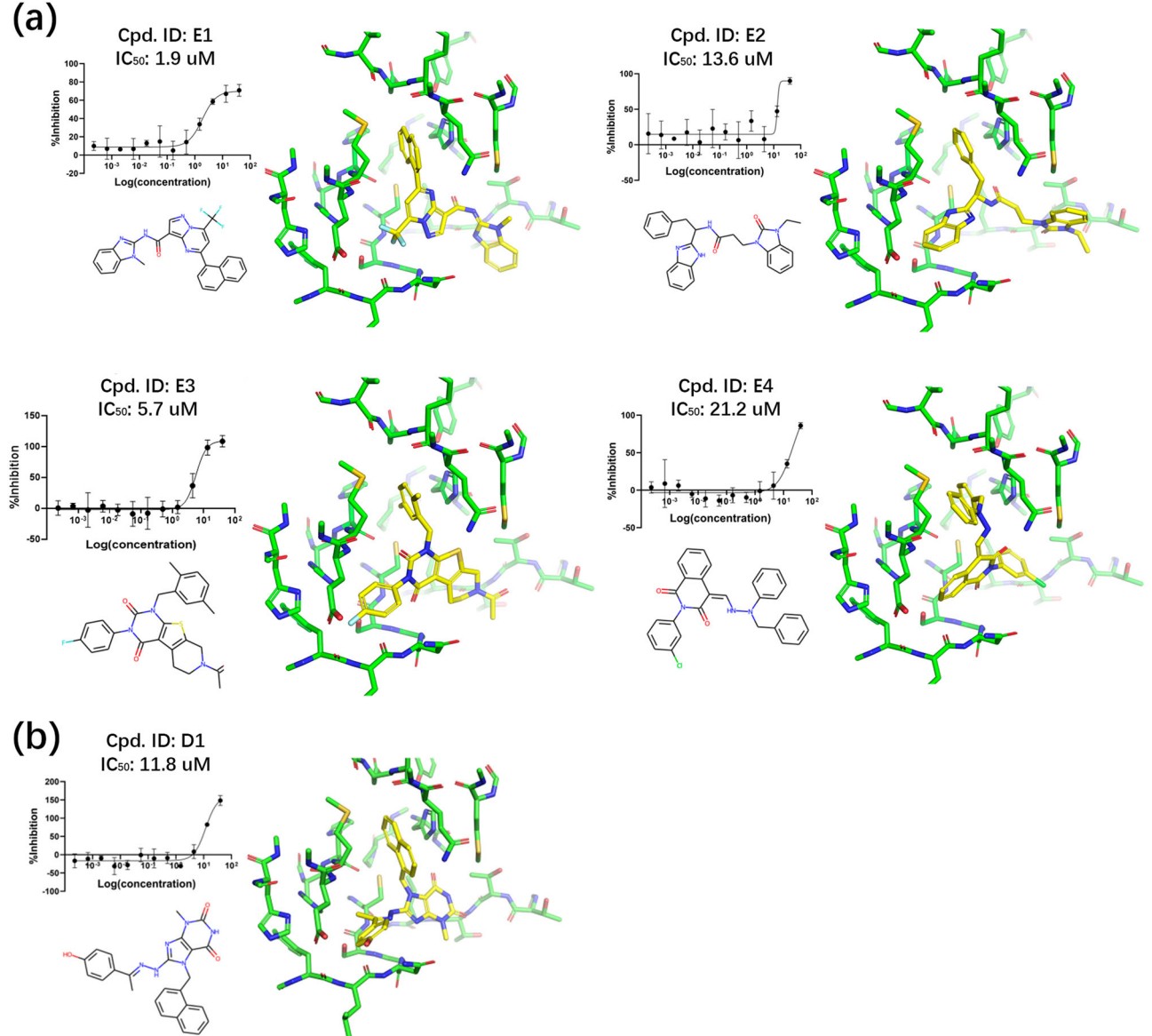

**Fig. 6 Active inhibitors of Covid-19 3CLpro. a** Active inhibitors among top 24 compounds ranked by considering both docking score and electron density-based grid matching score (ExptGMS). **b** Active inhibitors among top 24 compounds ranked based on docking score alone. Docking poses are shown as sticks, with ligands colored in yellow and pockets colored in green. Error bars represent the standard deviation from three replications.

In summary, our research highlights the importance of data, unlike other computational approaches that focus on algorithms, and demonstrates the value of experimental ED in enriching active compounds in virtual screening. We hope that our study will contribute to the community as a novel data source and open a new door for future algorithmic studies.

## Methods

**Datasets**. The DUD-E[15] dataset was used in this study. After removing the targets lacking a qualified ED map in the PDB, 85 targets were retained. GlideSP[22] docking poses and scores of all active compounds and decoys cited in DUD-E, for these 85 targets, were obtained from a previous study[16].

**Experimental ED map preparation and ExptGMS grid generation**. The coordinates and map coefficients were downloaded from the RCSB PDB web server[23]. The sigma (σ)-scaled $2F_o$–$F_c$ maps were synthesized at specific resolutions using Phenix[24], to cover the ligands and a 5 Å region around them. Specifically, the highest available resolution X-ray diffraction data in *.mtz* format were used as input for *phenix.fft* to generate electron density maps at various resolutions, which were specified by the parameter *d_min*. To create an ExptGMS grid based on an ED map, the map was first discretized into grids with a 0.3 Å interval. The value assigned to a grid point was the $2F_o$–$F_c$ ED intensity at that particular point. Grid points within the van der Waals radius of the pocket residue atoms were removed. Grid points with values of less than 0 σ were also removed. All the experimental maps involved in this study are electron density maps obtained through X-ray crystallography. No Coulomb potential maps from Cryo-EM were involved.

**Alpha sphere-based pocket shape preparation and grid generation**. An alpha sphere-based pocket shape was produced using FPocket version 4.0[25]. To generate a grid based on pocket shape, the region filled with alpha spheres was covered with grid points at intervals of 0.3 Å. Each grid point was assigned a value of one.

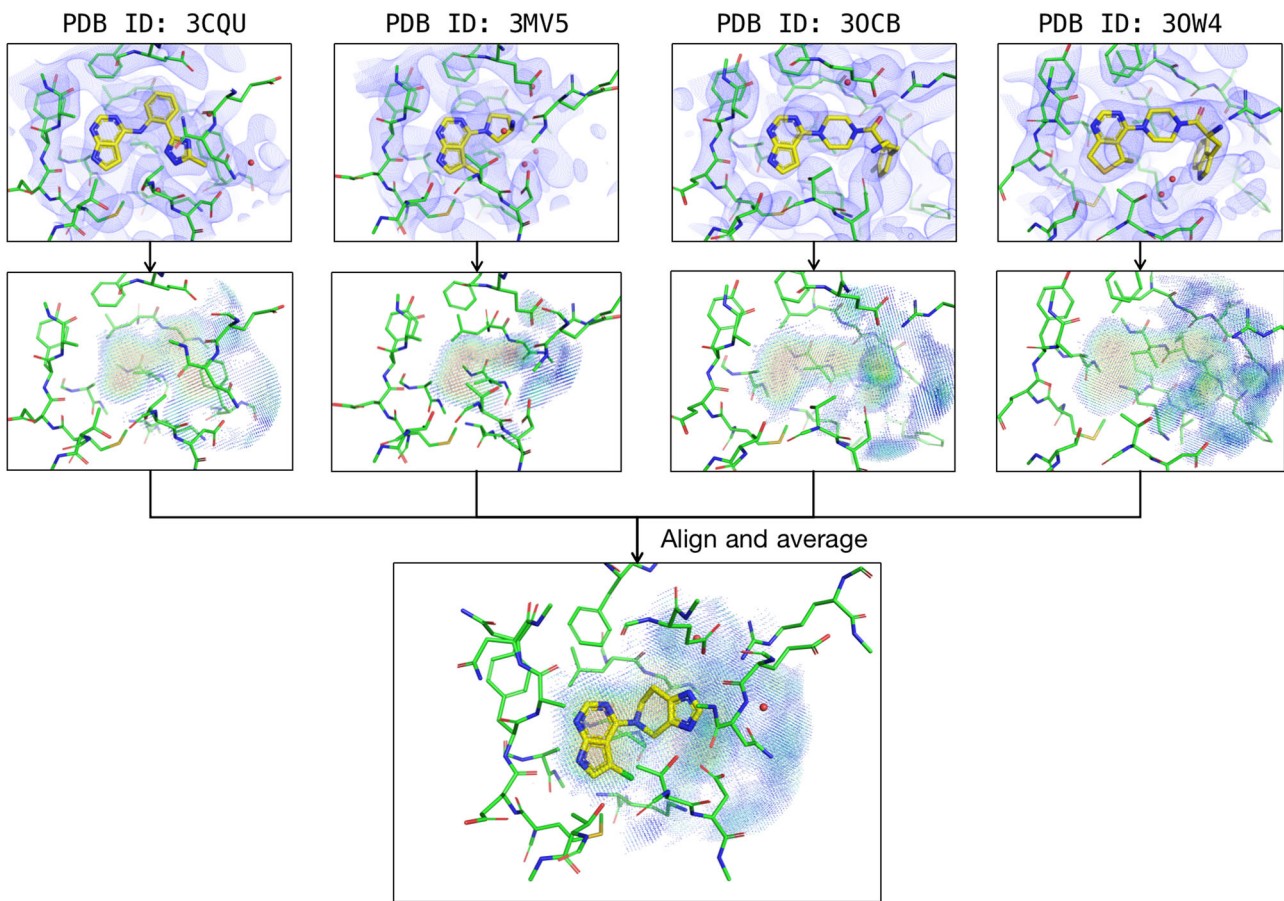

**Fig. 7 Construction of multi-crystal averaged ExptGMS grid for AKT1.** ED maps of four crystal structures of AKT1 (3CQU, 3MV5, 3OCB, 3OW4) are used to create multi-crystal averaged ExptGMS.

**Table 2 Performance of multi-crystal averaged ExptGMS vs. single-crystal ExptGMS.**

| Method[*a] | Number of active compounds among the top $N$ ranked[*b] | | | | |
|---|---|---|---|---|---|
| | $N = 10$ | $N = 50$ | $N = 100$ | $N = 500$ | $N = 1000$ |
| Averaged ExptGMS (3CQU, 3MV5, 3OCB, 3OW4) | 4 | 8 | 11 | 27 | 49 |
| ExptGMS (3CQU) | 0 | 3 | 6 | 14 | 20 |
| ExptGMS (3OW4) | 1 | 1 | 5 | 22 | 49 |
| ExptGMS (3MV5) | 3 | 7 | 7 | 16 | 24 |
| ExptGMS (3OCB) | 2 | 3 | 9 | 26 | 46 |

[*a]ExptGMS grids in this table are created based on 3 Å resolution EDs;
[*b]The ranking is solely based on ExptGMS score.

Grid points within the van der Waals radius of the pocket residue atoms were removed.

**ExptGMS scoring function**. A scoring function was designed to measure the degree of matching between the ligand conformations and ExptGMS grids [Eq. (1)].

$$\text{MatchScore} = S_{vac} - S_{occ} + P \tag{1}$$

where $S_{vac}$ represents contribution of vacant grid points that have intensity values, but are not occupied by any ligand atoms; $S_{occ}$ represents contribution of grid points occupied by ligand atoms; and P represents contribution of ligand atoms with no nearby grid points. A smaller MatchScore indicates a better match.

$S_{occ}$ is defined using Eq. (2):

$$S_{occ} = \sum_{m \in M} \sum_{\mathbf{r} \in R_m} w_m \rho(\mathbf{r}) \tag{2}$$

where $M$ represents all heavy atoms in the ligand; $R_m$ represents grid points located within a radius of 0.4 Å around a given atom $m$; $\rho(r)$ indicates the intensity value at grid point $r$; and $w_m$ represents the electron number of atom $m$ with a ceiling value of 9.

$S_{vac}$ is defined using Eq. (3):

$$S_{vac} = \sum_{\mathbf{v} \in V} \rho(\mathbf{v}) \tag{3}$$

where $V$ indicates vacant grid points with no ligand atoms within a radius of 0.4 Å; and $\rho(v)$ represents the intensity value of grid point $v$.

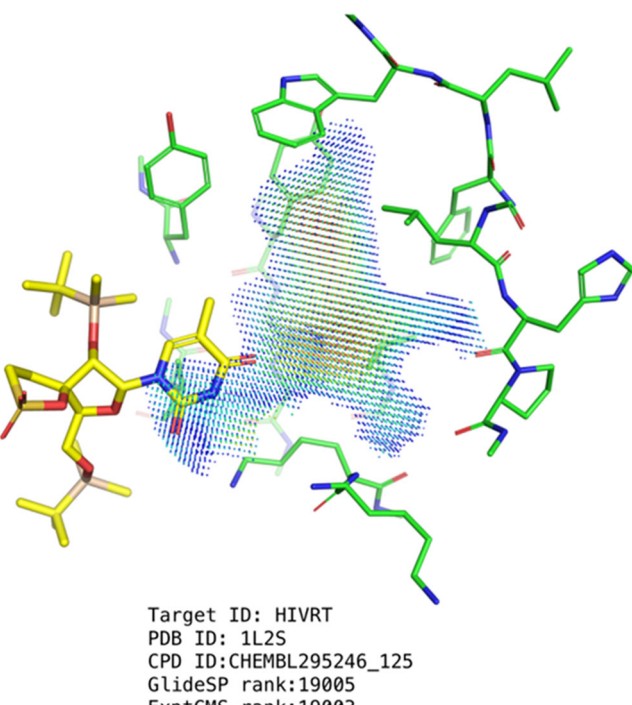

Target ID: HIVRT
PDB ID: 1L2S
CPD ID:CHEMBL295246_125
GlideSP rank:19005
ExptGMS rank:19003

**Fig. 8 Example showing the necessity of applying ExptGMS-based binding-pose searching.** The active compound colored in yellow is unsuccessful in docking to the designated pocket.

P is defined using Eq. (4):

$$P = n_{out}\left(\frac{S_{occ}}{n_{in}}\right) \quad (4)$$

where $n_{in}$ and $n_{out}$ denote the number of ligand atoms with and without grid points found within a radius of 0.4 Å, respectively.

**Molecule similarity and diversity**. The Tanimoto index was used to measure the 2D similarity between two molecules based on their ECFP4 fingerprints. The diversity of a set of molecules was defined as:

$$Diversity = 1 - (average\ pairwise\ 2D\ similarity\ among\ the\ molecules) \quad (5)$$

To measure 3D similarity, the Manhattan distance between the two molecules was calculated using their USRCAT descriptors[17]. ECFP4 and USRCAT calculations were performed using RDKit[26].

**MM/GBSA**. The receptor structure was prepared using the Protein Preparation Wizard program. To calculate the single point MM/GBSA binding free energy of the ligand–receptor complex, we used the Prime program. All residues within 4 Å of the ligand were treated as flexible during minimization. The Protein Preparation Wizard and Prime programs used in this study were sourced from the Schrödinger Suite (Release 2022-3). The force-file used is OPLS4.

**Machine-learning approach utilizing multi-resolution ExptGMS**. The DUD-E dataset was split into two separate subsets, a training set containing 73 targets, and a test set containing 12 targets (Supplementary Table S1). The sequence identities between target proteins were calculated using the Basic Local Alignment Search Tool (BLAST)[27] from NCBI. To avoid data leakage during the machine-learning process, it was ensured that none of the targets in the test set shared more than 30% sequence

identity with any sequence in the training set, thus minimizing the potential for sequence similarity bias.

A series of ExptGMS grids were generated using experimental ED maps of varying resolutions (2.5, 3.0, 3.5, 4.5, and 5.5 Å). For each ExptGMS grid, small molecules were scored using Eq. (1), and the scores were normalized to uniform features. Since the ExptGMS score may vary significantly among different targets, we employed by-target normalization in our approach. The mean and standard deviation of the ExptGMS scores for each target were calculated, and the original ExptGMS score was transformed to Z-score ($Z = (x − \mu)/\sigma$, where $\mu$ represents mean value and $\sigma$ represents standard deviation). Similarly, the alpha sphere matching, GlideSP, and MM/GBSA scores were also normalized using by-target Z-score, while the other features remained unchanged during our feature generation procedure.

In our GBDT model, ED scores from different resolutions were treated as a single feature with different preferences. Instead of using a statistical value such as the mean or median as a unique representation, we trained a group of decision trees by combining ExptGMS scores with other features. These submodels were further ensembled using the gradient boosting method. To address the issue of imbalanced positive and negative samples in the dataset, during training, a label-balancing strategy was introduced in which weights were assigned to different samples that were inversely proportional to their quantity. GBDT model was implemented using Scikit-learn[28] with parameters documented in Supporting Information (Supplementary Table S4 and Supplementary Fig. S4).

**Software for figures and tables**. All structures and ED figures were made using PyMOL. Analyses were performed using the Pandas[29], NumPy[30], and Scikit-learn[28]. Scatter plots were constructed using Matplotlib[31] and Seaborn[32] libraries.

**Covid-19 3CLpro-inhibitor virtual screening and biochemical assay**. The pocket used for virtual screening was obtained from SARS-CoV-2 3CL protease (PDB ID 7VU6) and prepared using Protein Preparation Wizard program. To conduct the screening, docking was performed on Glide program, against an in-house virtual compound library containing the structures of more than 8 million commercially accessible compounds. During screening, constraints were set so that the output ligand would require to form at least one hydrogen bond with the amide group of G143 or E166 in the pocket.

After docking, compounds were ranked based on their GlideSP score. The top 100,000 compounds were then subjected to an ExptGMS score calculation using a 3.0 Å ED map. By intersecting the top 500 compounds ranked by ExptGMS score with the top 1000 compounds ranked by docking score, we selected 24 molecules. These 24 compounds were evaluated using wet-lab tests to determine their inhibitory rates and IC50 values. As controls, an additional 24 compounds ranked solely by docking scores were also tested.

SARS-COV-2 3 CLpro (EC: 3.4.22.69) is a 3C-like proteinase that recognizes substrates containing the core sequence [ILMVF]-Q-↓-[SGACN][33,34]. The inhibition potency of a potential inhibitor was determined by FRET-based assay using a FRET-compatible peptide substrate MCA-AVLQ↓SGFR-Lys (Dnp)-Lys-$NH_2$ ("↓" indicates the cleavage site). MCA fluorescence is initially quenched by the Dnp group until cleavage (at the cleavage site) separates them. The maximum excitation light of MCA is 320 nm, while the maximum emission wavelength is 405 nm. The activity of 3CLpro was detected by measuring fluorescence. The protease inhibition rates of the compounds were measured as follows: each reaction mixture contains 0.15 μM 3CLpro (having a P132H mutation, $3CLpro^{P132H}$) and 40 μM inhibitor in 120 μL total volume in 96-well black polystyrene, flat bottom plates (Labselect, China). For

IC$_{50}$ determination, the reaction mixtures had 0.15 μM 3CLpro$^{P132H}$ and different concentrations of inhibitors in 120 μL total volume. 3CLpro$^{P132H}$ was preincubated with the compound for 30 min at room temperature. Subsequently, the fluorescence resonance energy transfer (FRET)-compatible peptide substrate MCA-AVLQSGFR-Lys(Dnp)-Lys-NH$_2$ was added to the reaction mixtures to initiate the reaction. Fluorescence was recorded for 20 min using 340 nm excitation and 405 nm emission filters at 10 s intervals on a multimode microplate reader (Thermo Scientific$^{TM}$ Varioskn$^{TM}$ LUX). The IC$_{50}$ values were determined by curve fitting, using a four-parameter equation in GraphPad Prism 8 software.

**Reporting summary**. Further information on research design is available in the Nature Portfolio Reporting Summary linked to this article.

## Data availability
The data analyzed in this study is included in this published article and its Supplementary Information files. Three additional supplementary data files were provided, with Supplementary Data 1 containing more details about Fig. 2, and Supplementary Data 2 and 3 containing more details about Table 1. The partial codes are available from the corresponding author upon reasonable request. Our model provides a service for academic users at https://exptgms.stonewise.cn/#/create.

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

## Acknowledgements
This study was funded by the National Key R&D Program of China (grant number 2022YFF1203004). This study was also supported by the Emergency Key Program of Guangzhou Laboratory (Grant No. EKPG21-30-2) and R&D Program of Guangzhou Laboratory (Grant No. SRPG22-011). This work was also supported by the Beijing Municipal Science and Technology Commission (No. Z211100003521001).

## Author contributions
B.H. and H.Z. conceived the study. B.H. provided instructions for all experiments. W.M. processed the ED map and conducted ExptGMS construction and evaluation. W.Z. conducted the biochemical tests. Y.L. developed the GBDT model. X.S. developed the scoring function. Y.X., Q.X. and W.Z. conducted the 3CLpro virtual screening. Y.D. and X.W. built online services. W.P. provided instructions for the biochemical assays. W.Z. provided instructions for developing the machine-learning model.

## Competing interests
The authors declare no competing interests.
