## [Peer Review File · Communications Chemistry]

Reviewers' comments:

Reviewer #1 (Remarks to the Author):

In this work, the authors use experimental densities to score docked poses. This approach is called ExptGMS and it rewards docked poses that overlap with the experimental density. Unfulfilled density is penalized, and docked atoms in low density regions are also penalized. ExptGMS does not influence docking as it re-scores poses that have been already docked. ExptGMS scores are combined in various ways with docking scores to rank docked molecules, and then performance is evaluated by the enrichment of active molecules among the top ranking molecules. The main claim of this paper is that ExptGMS combined with Glide scores leads to larger enrichment than Glide scores alone. Due to the size of the datasets where this was observed, I think this conclusion is partially supported by the data. I'll elaborate on that below. This paper is certainly of interest for those who do method development, and there may be a few drug discovery targets that may benefit from ExptGMS as presented herein, but users would have to try to understand whether or not ExptGMS is adequate for their system.

There are essentially three comparisons of the enrichment between Glide and Glide+ExptGMS. Each of these used a different algorithm to create the combined Glide+ExptGMS score.

The first comparison is in Figure 2 where 85 targets from DUD-e were used. The combined score is the Glide score but considering only poses that fell in the top 10% of ExptGMS scores. Here, the performance of Glide and Glide+ExptGMS is similar. It's nice that this is reported. Notably, the performance of ExptGMS alone is not much worse than that of Glide alone.

The second comparison uses a machine learning method, gradient boosting decision tree (GBDT), to combine ExptGMS and Glide scores. Importantly, there are multiple ExptGMS scores from densities at different resolutions, and not all targets have densities at each resolution. The enrichment from the machine learned score is about 20 % larger than the Glide score alone. The test set consists of 13 systems from the DUD-e. Typically, the enrichment varies widely from system to system and across scoring methods. For this reason, it would be valuable to report the enrichment for each of the systems. Likely, Glide+ExptGMS performs better in some targets and worse in others, and it can be valuable to inspect each target and try to understand what are the characteristics that make ExptGMS beneficial or detrimental to enrichment. Furthermore, with only 13 systems in the test set, it would be nice to have confidence intervals (e.g., by bootstrapping) on the enrichment values.

The third comparison is a prospective virtual screen of 8 million molecules in which 24 were selected based on Glide scores and 24 from Glide+ExptGMS scores. At single concentration, nine molecules selected from ExptGMS+Glide inhibited 3CLpro at least by 50%, while only 3

selected from Glide scores exhibited the same inhibition. However, for about half of the molecules, the dose response curves did not confirm the activity at single concentration, so there's some uncertainty in the data, as is normal for single concentration data. For the combined Glide+ExptGMS score, the selected molecules were in top 1000 from Glide and top 500 ExptGMS scores. Thus, the combined score is specialized for this virtual screen. For other systems it is possible that the top molecules from one score are also well ranked by the other score, or the intersection between both sets of top molecules is empty.

Since three different ways to combine ExptGMS and Glide were used, I can imagine that finding a simple way to incorporate experimental densities into docking scores that is applicable to every system is probably very difficult, and deploying it for prospective drug discovery will require some form of manual intervention.

Some more technical comments and suggestions. Reproducibility is compromised without many of these.

1. Report the PDB IDs used for docking and for the density of ExptGMS for each DUD-E target
2. Report enrichment for each DUD-E target, both for Table 1 and Fig 2
3. Detail the final parameters (and/or hyperparameters) of the GDBT
4. Report performance on the training set for GDBT, and explain how the parameter tuning using cross validation was carried out. How was the training set divided and which parameters were most important. This information can help in understanding how general the model is. Overfitting is always a concern with machine learning.
5. Report confidence intervals, especially for Table 1
6. Figure 2 is a nice way to compare various re-scoring methods using both the enrichment of actives and their chemical diversity. I particularly like it
7. State whether or not any Cryo-EM densities were used and for which systems
8. What software was used for GDBT
9. Some more MMGBSA details would be nice, especially the force field version and whether it was single point or MD simulation
10. State the size of the test set (n=13) in the caption of Table 1

11. Regarding the 3CLpro screen, there is no mention of visual inspection. I would recommend stating that visual inspection, or any form of manual selection, was not used (assuming that really is the case) as this is common practice.

12. line 104: excluding the region occupied by the pocket itself - does this mean protein atoms?

13. Since only 1/3 of the DUD has all resolutions (line 212) what are the GBDT descriptors when a target doesn't have the full range of resolutions?

Diogo Santos-Martins

Reviewer #2 (Remarks to the Author):

Virtual screening (VS) has emerged as a powerful method for quickly screening vast libraries of compounds and reducing them down to a small pool of candidates that can be investigated in a time and cost efficient manner. While great improvements have been made in both reducing the computational time as well as the biophysical modeling of ligand binding, the success rate of VS can still vary between targets and limitations exist. Traditional virtual screening methods make simplifying assumptions about empirical parameters and consequently VS may not capture the intricacies of molecular recognition. These scoring function simplifications lead to inaccuracies in predicting receptor-ligand poses and relative affinities. In this paper, the authors address whether incorporating experimental electron densities (ED) of ligand-bound structures can correct for these simplifications in the scoring function and therefore improve enrichment and diversity of compounds in VS. Based on a pre-existing crystal structure of a receptor, the authors calculate the intensity of ED at grid points and incorporate those values into a modified scoring scheme which they call ExptGMS. Incorporating ED of the binding site also introduces ED of solvent molecules and alternative side chain conformations which provides information about pocket dynamics and accessibility that can be relevant to ligand binding. Such information might implicitly encode the displacement of water molecules, rearrangement of side chains upon ligand binding or other features. These aspects may be simplified (rigid structures) or omitted (solvent) in other VS programs.

To test the effectiveness of their approach, they benchmarked their method against existing ligand or receptor-based VS software showing a small improvement in balancing positive hit identification and hit diversity. The authors then optimized their algorithm by factoring in a low resolution cutoff to the ED. High resolution ED will result in high intensity grid points; because the authors scoring function favors ligands that occupy ED peaks this could lead to bias against compounds that miss the exact centers of those peaks. Therefore, the authors also included low pass filtered (maps calculated only using low resolution reflections) data which produces a

more smoothed distribution of ED intensity which they argue accounts for more conformational variability, however, this must be balanced against using too low resolution data, which would result in information loss about ligand fitting to features in the density volume.

They then tested it against the SARS CoV 2 protease 3CLpro which yielded several compounds that exhibited IC50's in the low micromolar range.

The major strength of this paper is a protocol for incorporating ED of bound ligands to alter the scoring function used in specific receptor binding sites in VS. Currently, this procedure produces comparable results to existing VS software such as GlideSP, which do not have any ED term.

Many existing docking algorithms work iteratively by gradually introducing more complex terms; they are aided by scoring functions which assign greater weight to molecules that can form empirical interactions with the receptor. In some cases, a ligand is missed either because the correct binding pose can't be found or because the scoring function penalizes a lack of strong interactions. In these cases, the ED based scoring function could be utilized to aid the scoring function. The inclusion of the solvent ED in mapping out the binding site and aiding in ligand placement to be particularly interesting from the perspective of developing VS docking tools that take advantage of apo-structures that don't have ligands already bound to them, which could expand the use of this method. The major weakness of this paper is, as mentioned by the authors, that this method is limited to proteins that have electron density with ligands bounds which limits its utility for receptors with no known ligand bound ED (although see point above for potential for expanding the domain of applicability). Given the current implementation of ExptGMS, we are curious to know if the authors tried generating electron density grids based on the solvent density in the binding site alone. Augmenting VS scoring functions by incorporating experimental ED may further improve docking scores by aiding the placement of molecules using existing binding data of ligands as well as solvent; however, currently the performance improvements offered by this method are modest.

Major points:

In figure 2, the authors compare how well ExptGMS performs relative to other virtual docking programs by examining the number of top 10, 50, and 100 compounds that are detected and the diversity of compounds. The 2D plot used does not sufficiently describe how differences between datasets used in the DUD-E database could affect these results. For example, did ExptGMS (and the other programs) do much better with some datasets than others? It would be helpful if the authors could show representative graphs from individual datasets to make the differences in performance more clear.

In figure 2 the authors benchmark ExptGMS against several VS software programs that are either ligand or receptor based. They note in figure 3 that their method can improve false negative/positive hit detection. What was the overall false positive/negative rates between the methods used? Do the authors see a relationship between the number of top 10, 50, and 100 compounds and the false negative/positive rate? Does this depend on the choice of software (ex ligand vs receptor based VS)? It would be helpful if the authors could further elaborate on

the false positive and negative hit detection rates between their method and the existing VS methods.

We feel the 3CLpro in vitro assay description is unclear as it is written. Is this a peptide displacement assay? Further, what was the construct for the peptide/choices of fluorophores? Further explanation of how the in vitro assay was constructed and performed would be helpful.

In Figure 5, the authors describe one possibility for why ExptGMS with different resolutions complemented each other by using solvent exposure in the pocket. How are t-test results in box plots? We feel that it is difficult to judge whether the red and blue boxes in each resolution have statistically significant differences.

In Table 1, the authors try to demonstrate the usefulness of multi-resolution analysis using a machine learning model. The table seems to show the advantage of the combination between GlideSP and multi-resolution ExptGSM. However, we are concerned about multiple hypothesis testing. This is also related to Figure 5. The authors tested more and more things about multi-resolution analysis but did not show a principled test. It would be helpful to get a deeper understanding about multi-resolution analysis if the authors could provide their thoughts and tests related to the principle.

Minor points:

Supplementary materials are mentioned but not available under biorxiv posting.

Reviewed by CJ San Felipe, Hiroki Yamamura, and James Fraser (UCSF)

Reviewers' comments:

Reviewer #1 (Remarks to the Author):

In this work, the authors use experimental densities to score docked poses. This approach is called ExptGMS and it rewards docked poses that overlap with the experimental density. Unfulfilled density is penalized, and docked atoms in low density regions are also penalized. ExptGMS does not influence docking as it re-scores poses that have been already docked. ExptGMS scores are combined in various ways with docking scores to rank docked molecules, and then performance is evaluated by the enrichment of active molecules among the top ranking molecules. The main claim of this paper is that ExptGMS combined with Glide scores leads to larger enrichment than Glide scores alone. Due to the size of the datasets where this was observed, I think this conclusion is partially supported by the data. I'll elaborate on that below. This paper is certainly of interest for those who do method development, and there may be a few drug discovery targets that may benefit from ExptGMS as presented herein, but users would have to try to understand whether or not ExptGMS is adequate for their system.

Reply:

We are grateful to you for appreciating our work and providing valuable feedback. We have revised the manuscript as per your suggestions. The pointwise response to your comments is provided below.

There are essentially three comparisons of the enrichment between Glide and Glide+ExptGMS. Each of these used a different algorithm to create the combined Glide+ExptGMS score.

The first comparison is in Figure 2 where 85 targets from DUD-e were used. The combined score is the Glide score but considering only poses that fell in the top 10% of ExptGMS scores. Here, the performance of Glide and Glide+ExptGMS is similar. It's nice that this is reported. Notably, the performance of ExptGMS alone is not much worse than that of Glide alone.

The second comparison uses a machine learning method, gradient boosting decision tree (GBDT), to combine ExptGMS and Glide scores. Importantly, there are multiple ExptGMS scores from densities at different resolutions, and not all targets have densities at each resolution. The enrichment from the machine learned score is about 20 % larger than the Glide score alone. The test set consists of 13 systems from the DUD-e. Typically, the enrichment varies widely from system to system and across scoring methods. For this reason, it would be valuable to report the enrichment for each of the systems. Likely, Glide+ExptGMS performs

better in some targets and worse in others, and it can be valuable to inspect each target and try to understand what are the characteristics that make ExptGMS beneficial or detrimental to enrichment. Furthermore, with only 13 systems in the test set, it would be nice to have confidence intervals (e.g., by bootstrapping) on the enrichment values.

Response: We appreciate the reviewer's suggestions.

We have used the bootstrapping strategy and provided confidence intervals on enrichment values in Table 1. in the revised manuscript. Details are given below in response to the question #5.

The third comparison is a prospective virtual screen of 8 million molecules in which 24 were selected based on Glide scores and 24 from Glide+ExptGMS scores. At single concentration, nine molecules selected from ExptGMS+Glide inhibited 3CLpro at least by 50%, while only 3 selected from Glide scores exhibited the same inhibition. However, for about half of the molecules, the dose response curves did not confirm the activity at single concentration, so there's some uncertainty in the data, as is normal for single concentration data. For the combined Glide+ExptGMS score, the selected molecules were in top 1000 from Glide and top 500 ExptGMS scores. Thus, the combined score is specialized for this virtual screen. For other systems it is possible that the top molecules from one score are also well ranked by the other score, or the intersection between both sets of top molecules is empty.

Since three different ways to combine ExptGMS and Glide were used, I can imagine that finding a simple way to incorporate experimental densities into docking scores that is applicable to every system is probably very difficult, and deploying it for prospective drug discovery will require some form of manual intervention.

Response: We appreciate the reviewer's informative and constructive suggestions for improving the manuscript and the ExptGMS method.

Some more technical comments and suggestions. Reproducibility is compromised without many of these.

1. Report the PDB IDs used for docking and for the density of ExptGMS for each DUD-E target

Response: We thank the reviewer for raising this point.

We have added such information to Table S1 in the revised Supporting Information.

Original

Table S1. List of targets in training set and test set

Targets in Training Set	Targets in Test set
AA2AR, ABL1, ACE, ACES, ADA, ADRB1, ADRB2, AKT1, AKT2, ALDR, ANDR, BRAF, CAH2, CASP3, CDK2, CP2C9, CP3A4, CSF1R, CXCR4, DRD3, DYR, EGFR, ESR1, ESR2, FA10, FA7, FABP4, FAK1, GCR, GLCM, GRIA2, HDAC2, HDAC8, HIVINT, HIVPR, HIVRT, HMDH, HS90A, HKK4, IGF1R, ITAL, JAK2, KIT, KITH, LKHA4, MAPK2, MET, MK01, MK10, MK14, MP2K1, NRAM, PARP1, PGH1, PGH2, PLK1, PPARA, PPARD, PPARG, PRGR, PTN1, PUR2, PYRD, RENI, ROCK1, SAHH, SRC, TGF1R1, THRB, TRYBI, UROK, VGFR2, WEE1	AMPC, AOFB, BACE1, DH11, DPP4, FNTA, INHA, KIF11, PA2GA, PDE5A, PNP, XIAP

Revised

Table S1. List of targets in training set and test set (N_{training}=73, N_{test}=12)

Target	PDBID	Training or Test	Type
AA2AR	3EML	Training	GPCR
ABL1	2HZI	Training	Kinase
ACE	3BKL	Training	Protease
... 85 lines			
PDE5A	1UDT	Test	Other Enzymes
PNPH	3BGS	Test	Other Enzymes
XIAP	3HL5	Test	Miscellaneous

All 85 targets with PDB ID are listed in the revised Table S1.

2. Report enrichment for each DUD-E target, both for Table 1 and Fig 2

Response: We thank the reviewer for raising this point.

We have provided two additional supporting files containing target enrichment information in the revised version.

Table1_Enrichment_by_target.csv (36 lines)

Target	PDBID	TopN	GlideSP	GlideSP and MM/GBSA	GlideSP and Alpha sphere	GlideSP and USRCAT	GlideSP and multi-resolution ExptGMS	GlideSP and TF3P
AMPC	1I2S	10	1	1	1	1	2	0
AOFB	1S3B	10	1	1	1	1.2	0	0
BACE1	3L5D	10	4	0	2	2	7	2
DH11	3FRJ	10	2	0	0	1	1	0
DPP4	2I78	10	8	7	8	9	10	9
FNTA	3E37	10	3	3	2	2	1	2
INHA	4TRJ	10	2	2	2	2	0	6
KIF11	3CJO	10	5	9	10	8	10	9
PA2GA	1KVO	10	7	8	8.2	8	5	10
PDE5A	1UDT	10	0	0.3	1.8	1.7	10	9

Fig2_Enrichment_by_target.csv (255 lines)

Target	PDBID	TopN	ExptGMS	GlideSP	GlideSP+ExptGMS	alpha sphere (pocket shape)	MM/GBSA	USRCAT (3D similarity)	ECFP (2D similarity)	ExptGMS shape-only
AA2AR	3EML	10	10	4	8	0	1	4	10	6
ABL1	2HZI	10	8	9	9	2	1	1	8	3
ACE	3BKL	10	9	9	9	7	2	6	10	9
ACES	1E66	10	2	3	0	0	3	5	8	0
ADA	2E1W	10	7	1	5	0	0	10	10	3
ADRB1	2VT4	10	10	6	8	0	2	2	10	3
ADRB2	3NY8	10	7	7	8	0	0	4	8	5
AKT1	3CQW	10	2	6	7	1	0	4	10	1
AKT2	3DOE	10	10	6	7	0	5	5	10	7
ALDR	2HV5	10	9	10	10	4	1	0	10	7
AMPC	1L2S	10	2	1	0	0	0	1	10	0
ANDR	2AM9	10	1	8	8	3	0	10	10	1
AOFB	1S3B	10	0	1	1	0	2	2	9	0

3. Detail the final parameters (and/or hyperparameters) of the GDBT

Response: We thank the reviewer for raising this point.

Key Parameters:

max_iter = 500

class_weight='balanced'

interaction_cst is restraint such that ED with different resolution will not interact with each other, with details are listed below:

```

'ds_ed_only': {'cols': ['normed_glide_score',
                      'normed_ed25',
                      'normed_ed30',
                      'normed_ed35',
                      'normed_ed45',
                      'normed_ed55',
                      ],
              'interaction_cst': [set(comb) for comb in itertools.product([0], list(range(1, 6)))]
              },
'ds_tf3p': {'cols': ['normed_glide_score',
                    'normed_tf3p',
                    ],
           'interaction_cst': [[0, 1]]
           },
'ds_mmgbsa': {'cols': ['normed_glide_score',
                      'normed_mmgbsa',
                      ],
              'interaction_cst': [[0, 1]]
              },
'ds_fpocket': {'cols': ['normed_glide_score',
                       'normed_fpocket_ed',
                       ],
               'interaction_cst': [[0, 1]]
               },
'ds_usrcat': {'cols': ['normed_glide_score',
                      'usrcat',
                      ],
              'interaction_cst': [[0, 1]]
              },

```

The rest parameters are default values:

learning_rate = 0.1

max_leaf_nodes = 31

max_depth=None

min_samples_leaf=20

l2_regularization=0

max_bins=255

warm_start=False

n_iter_no_change=10

tol=1e-7

The above listed details were included in

“Table S4. Details of GBDT training parameters” in the revised Supporting Information.

Table S4. Details of GBDT training parameters

Key Parameters	Other Parameters
max_iter = 500 class_weight='balanced' interaction_cst is restraint such that ED with different resolution will not interact with each other, with details listed below: <pre> 'ds_ed_only': {'cols': ['normed_glide_score', 'normed_ed25', 'normed_ed30', 'normed_ed35', 'normed_ed45', 'normed_ed55',], 'interaction_cst': [set(comb for comb in itertools.product([0, 1], list(range(1, 6))))], 'ds_tf3p': {'cols': ['normed_glide_score', 'normed_tf3p'], 'interaction_cst': [[0, 1]] }, 'ds_mmgbsa': {'cols': ['normed_glide_score', 'normed_mmgbsa'], 'interaction_cst': [[0, 1]] }, 'ds_fpocket': {'cols': ['normed_glide_score', 'normed_fpocket_ed'], 'interaction_cst': [[0, 1]] }, 'ds_usrcat': {'cols': ['normed_glide_score', 'usrcat'], 'interaction_cst': [[0, 1]] }, </pre>	<pre> learning_rate = 0.1 max_leaf_nodes = 31 max_depth=None min_samples_leaf=20 l2_regularization=0 max_bins=255 warm_start=False n_iter_no_change=10 tol=1e-7 </pre>

4. Report performance on the training set for GBDT, and explain how the parameter tuning using cross validation was carried out. How was the training set divided and which parameters were most important. This information can help in understanding how general the model is. Overfitting is always a concern with machine learning.

Response: We appreciate the reviewer's comments.

The performance of GBDT on the training set is shown below, through a plot displaying the loss versus iterations. To prevent overfitting, a cross-validation strategy was employed during the training process. This involved using a validation set, which consisted of 10% of the training set data. The cross-validation set was used for early stopping in order to prevent overfitting. The training of GBDT was initially set to iterate for 500 steps. As shown in the figure below, it stopped at step 63. During the training process, it is observed that the validation loss initially decreases and subsequently stabilizes, without showing any signs of an upward trend. This pattern suggests that the model does not suffer from overfitting.

This plot was also included in revised Supporting Information as “Figure S4. Training curve of GBDT (GlideSP and multi-resolution ExptGMS)”.

As mentioned by the reviewer, hyperparameter tuning can potentially lead to overfitting while increasing performance on certain test datasets. In our work, most of the hyperparameters were set to their default values, and no hyperparameter tuning was performed. We made two changes to address the dataset size and label imbalance issue: we changed `max_iter` to 500 and `class_weight` to 'balanced'. Specifically, setting the `class_weight` to "balanced" allows for the adjustment of the weight of a sample in inverse proportion to its class (i.e., label) frequency in the input data. In addition, we configure the feature interaction parameter 'interaction_cst' by specifying a list of sets. This prevents ED with different resolution from being included in the same tree. The details of 'interaction_cst' can be found in Table S4 of the revised Supporting Information.

5. Report confidence intervals, especially for Table 1
Response: We appreciate the reviewer’s suggestions.

In the revised manuscript (lines 308 to 314), a paragraph has been added to describe the utilization of a bootstrapping method for calculating a 90% confidence interval.

“...The confidence intervals for the active compounds in the top N were obtained using the bootstrapping method. Specifically, samples were randomly selected with replacement from the test dataset until the selected sample size matched the size of the test dataset. Considering all the selected samples, the average numbers of active compounds in the top 10, 50, and 100 results were calculated, respectively. By repeating this process 200 times, a distribution of results was generated. From this distribution, the mean value and percentile confidence interval were computed.”

We updated Table 1 by including the confidence intervals.

Table 1. Performance of GBDT models trained using different features on the DUD-E test set (N=12).

Features used in the model	Average number of Active Compound in Top N, with 90% Confidence Interval			Diversity ^{*b}
	N = 10	N = 50	N = 100	
GlideSP and multi-resolution ExptGMS ^{*a}	5.4 [5.0, 5.8]	21.6 [20.4, 22.9]	33.2 [30.9, 35.1]	0.64 [0.60, 0.67]
GlideSP and TF3P	5.4 [4.9, 5.8]	19.5 [18.1, 20.8]	27.7 [26.0, 29.6]	0.62 [0.60, 0.64]
GlideSP and MM/GBSA	4.5 [4.0, 5.2]	18.5 [17.0, 19.9]	30.1 [28.1, 32.2]	0.66 [0.62, 0.68]
GlideSP and USRCAT	4.5 [3.9, 5.1]	18.1 [16.9, 19.3]	30.4 [28.0, 32.3]	0.63 [0.59, 0.66]
GlideSP and Alpha sphere	5.1 [4.6, 5.6]	18.4 [17.0, 19.8]	29.2 [27.2, 31.0]	0.64 [0.61, 0.68]
GlideSP ^{*c}	4.3 [3.8, 4.8]	17.3 [16.1, 18.6]	29.9 [27.9, 31.9]	0.66 [0.62, 0.68]

a: Multiresolution indicates ExptGMS at resolutions of 2.5, 3.0, 3.5, 4.5, and 5.5 Å.

**b*: Diversity of active compounds among the top 100 ranked compounds.

**c*: Ranking was performed directly using the GlideSP score without using a trained model.

6. Figure 2 is a nice way to compare various re-scoring methods using both the enrichment

of actives and their chemical diversity. I particularly like it

Response: We are grateful to you for appreciating our work and providing valuable feedback.

7. State whether or not any Cryo-EM densities were used and for which systems

Response: We appreciate the reviewer's suggestions.

Because Cryo-EM experimental maps are Coulomb potential maps, which are different from electron density maps obtained by X-ray crystallography, our studies in this work only focused on electron density maps and did not involve any Cryo-EM densities. We are currently working on Cryo-EM based grids and will publish our findings in the near future.

To clarify the aforementioned point, we have included the following statement in the revised manuscript (lines 446-448).

"...All the experimental maps involved in this study are electron density maps obtained through X-ray crystallography. No Coulomb potential maps from Cryo-EM were involved."

8. What software was used for GBDT

" sklearn.ensemble.HistGradientBoostingClassifier" was used for GBDT implementation.

We have added the following statement in the revised manuscript (line 525).

*"...GBDT model was implemented using Scikit-learn¹ with parameters available in **Supporting Information Table S4 and Figure S4.**"*

9. Some more MMGBSA details would be nice, especially the force field version and whether it was single point or MD simulation

Reply: We appreciate the reviewer's suggestions.

MM/GBSA section has been updated in the revised manuscript (lines 493 to 498) to provide additional information.

"...The receptor structure was prepared using the Protein Preparation Wizard program. To calculate the single point MM/GBSA binding free energy of the ligand-receptor complex, we used the Prime program. All residues within a 4 Å of the ligand were treated as flexible during minimization. The Protein Preparation Wizard and Prime programs used in this study were sourced from the Schrödinger Suite (Release 2022-3). The force-file used is OPLS4."

10. State the size of the test set (n=13) in the caption of Table 1

Response: We appreciate the reviewer's suggestions.

In the revised manuscript (line 316), the caption of Table 1 has been updated as follows:

***Table 1.** Performance of GBDT models trained using different features on the DUD-E test set (N=12)."*

11. Regarding the 3CLpro screen, there is no mention of visual inspection. I would recommend stating that visual inspection, or any form of manual selection, was not used (assuming that really is the case) as this is common practice.

Response: We appreciate the reviewer's suggestions.

Our purpose was to test the performance of our computational methods. Therefore, in this case no visual inspection was used.

In the revised manuscript (line 334), below sentence has been added.

"...It is important to mention that no visual inspection or manual selection was involved in the selection of the aforementioned 48 compounds."

12. line 104: excluding the region occupied by the pocket itself - does this mean protein atoms?

Response: Thank you for the valuable comment

Yes. To improve clarity, we have updated the sentence in the revised manuscript (line 104).

"... The grids only cover the regions inside and around the pocket, excluding the region occupied by the pocket itself (i.e. protein atoms)."

13. Since only 1/3 of the DUD has all resolutions (line 212) what are the GBDT descriptors when a target doesn't have the full range of resolutions?

Response: We thank the reviewer for raising this point.

The writing in the original manuscript was unclear. What we intend to express is that "for 1/3 of the targets, ExptGMS outperforms GlideSP regardless of the resolution used," not that "only 1/3 of the targets have data at all five resolutions."

In the revised manuscript, we have elaborated such unclear expressions (lines 235–240).

"... To enhance clarity, we listed all 85 tested targets in a circle, and colored the targets using a resolution-specific color, if the active compound enrichment of ExptGMS+GlideSP at that resolution outperformed GlideSP. Figure 5a displays the union of these coloured targets across different resolutions, covering approximately 75% of the targets, while the intersection of these coloured targets accounts for only one-third of the total."

The unclear expression in the original manuscript raises concerns for the reviewer regarding the absence of multiresolution data for a specific target. Actually, this should not be a concern because of the following two points:

- 1. If a target has 2.5 Å resolution experimental ED data, then resolutions lower than 2.5 Å, such as 3.5 Å, 4.5 Å, and 5.5 Å, are also intrinsically available. So, we only concern about the upper limit of the resolution.*
- 2. We have summarized the distribution of resolution for the available structures*

in the table below. It shows that for PDBbind which contains most of the intensively studied drug discovery targets, over 95% of them have resolution upper limit hitting 3.0 Å (i.e. having 4 features for the GBDT model). Even for the entire PDB database, such a ratio is also above 85%.

Resolution	DUD-E (85 Entries)	PDBbind v.2020 (15,253 Entries)	RCSB PDB (192,408 Entries)
2.5	82.35%	78.83%	69.61%
3	97.65%	95.79%	86.27%
3.5	100.00%	99.44%	94.04%
4.5	100.00%	99.99%	98.26%
5.5	100.00%	100.00%	>98.41%

Regarding the GBDT descriptors, when a target lacks one or more resolution data, GBDT can accept a feature matrix with empty values. In this case, we leave the missing values unchanged. Specifically, if a target does not have the full range of resolutions, the values corresponding to those missing resolutions will be left blank.

Reviewer #2:

Virtual screening (VS) has emerged as a powerful method for quickly screening vast libraries of compounds and reducing them down to a small pool of candidates that can be investigated in a time and cost efficient manner. While great improvements have been made in both reducing the computational time as well as the biophysical modeling of ligand binding, the success rate of VS can still vary between targets and limitations exist. Traditional virtual screening methods make simplifying assumptions about empirical parameters and consequently VS may not capture the intricacies of molecular recognition. These scoring function simplifications lead to inaccuracies in predicting receptor-ligand poses and relative affinities. In this paper, the authors address whether incorporating experimental electron densities (ED) of ligand-bound structures can correct for these simplifications in the scoring function and therefore improve enrichment and diversity of compounds in VS. Based on a pre-existing crystal structure of a receptor, the authors calculate the intensity of ED at grid points and incorporate those values into a modified scoring scheme which they call ExptGMS. Incorporating ED of the binding site also introduces ED of solvent molecules and alternative side chain conformations which provides information about pocket dynamics and accessibility that can be relevant to ligand binding. Such information might implicitly encode the displacement of water molecules, rearrangement of side chains upon ligand binding or other features. These aspects may be simplified (rigid structures) or omitted (solvent) in other VS programs.

To test the effectiveness of their approach, they benchmarked their method against existing ligand or receptor-based VS software showing a small improvement in balancing positive hit identification and hit diversity. The authors then optimized their algorithm by factoring in a low resolution cutoff to the ED. High resolution ED will result in high intensity grid points; because the authors scoring function favors ligands that occupy ED peaks this could lead to bias against compounds that miss the exact centers of those peaks. Therefore, the authors also included low pass filtered (maps calculated only using low resolution reflections) data which produces a more smoothed distribution of ED intensity which they argue accounts for more conformational variability, however, this must be balanced against using too low resolution data, which would result in information loss about ligand fitting to features in the density volume.

They then tested it against the SARS CoV 2 protease 3CLpro which yielded several compounds that exhibited IC₅₀'s in the low micromolar range.

The major strength of this paper is a protocol for incorporating ED of bound ligands to alter the scoring function used in specific receptor binding sites in VS. Currently, this procedure produces comparable results to existing VS software such as GlideSP, which do not have any ED term. Many existing docking algorithms work iteratively by gradually introducing more complex terms; they are aided by scoring functions which assign greater weight to molecules that can

form empirical interactions with the receptor. In some cases, a ligand is missed either because the correct binding pose can't be found or because the scoring function penalizes a lack of strong interactions. In these cases, the ED based scoring function could be utilized to aid the scoring function. The inclusion of the solvent ED in mapping out the binding site and aiding in ligand placement to be particularly interesting from the perspective of developing VS docking tools that take advantage of apo-structures that don't have ligands already bound to them, which could expand the use of this method. The major weakness of this paper is, as mentioned by the authors, that this method is limited to proteins that have electron density with ligands bound which limits its utility for receptors with no known ligand bound ED (although see point above for potential for expanding the domain of applicability). Given the current implementation of ExptGMS, we are curious to know if the authors tried generating electron density grids based on the solvent density in the binding site alone.

Response: We are grateful to you for appreciating our work and providing valuable feedback. We have revised the manuscript as per your suggestions. The pointwise response to your comments is provided below.

It is interesting and important to extend the application of ExptGMS in apo structures where only solvent density in the binding site is available.

Usually, in addition to binding only with solvent molecules instead of molecule-sized binders, apo-structures differ from holo-structures in several other aspects. These include variations in volume, the hydrophobic ratio of the surface, and conformational changes of key residues. The dynamics of key residues, even the backbone, make it challenging to find the holo-pocket using the apo-structure as a starting point.

Multi-resolution ExptGMS, especially at low resolution, contains dynamic information about the pocket, making it a promising direction for study. We do observe promising results in enriching active compounds in HTVS in the following scenarios:

- 1. Holo-state pockets exhibit a small root mean square deviation (RMSD) when compared to apo structures.*
- 2. Close to scenario one, but there are some side chains of Arg and Lys residues that move with large temperature factors.*

We are currently working on more challenging cases involving the movements of protein loops. The related results will be published in the near future.

Augmenting VS scoring functions by incorporating experimental ED may further improve docking scores by aiding the placement of molecules using existing

binding data of ligands as well as solvent; however, currently the performance improvements offered by this method are modest.

Response: We appreciate the reviewer's informative and constructive suggestions for improving the manuscript and the ExptGMS method.

Major points:

1. In figure 2, the authors compare how well ExptGMS performs relative to other virtual docking programs by examining the number of top 10, 50, and 100 compounds that are detected and the diversity of compounds. The 2D plot used does not sufficiently describe how differences between datasets used in the DUD-E database could affect these results. For example, did ExptGMS (and the other programs) do much better with some datasets than others? It would be helpful if the authors could show representative graphs from individual datasets to make the differences in performance more clear.

Response: We appreciate the reviewer's suggestions.

To address the reviewer's concern, we divided the DUD-E database into eight datasets based on the target classes documented in the original DUD-E paper². The target classes include kinase, protease, nuclear receptor, GPCR, cytochrome P450, ion channel, other enzymes, and miscellaneous. The number of targets belonging to each target type is listed below.

Target Type	# of targets included
Other Enzymes	29
Kinase	23
Protease	12
Nuclear Receptor	8
GPCR	5
Miscellaneous	5
Cytochrome P450	2
Ion Channel	1

Given the limited target number for the bottom 5 ranking target types, we conducted the enrichment and diversity analysis using the 3.0 Å resolution ExptGMS for the top 3 ranking target types, as we did in Figure 2. The results were displayed in the figures below, which are also added as Figure S1 in the revised Supporting Information.

Figure S1. Comparison of ExptGMS with benchmark technologies on DUD-E dataset by target class.

Based on the above figure, it can be observed that when considering both enrichment and diversity, ExptGMS outperforms most of the benchmark approaches. More importantly, Glide+ExptGMS enriched a greater number of active compounds in the top 10 or top 50 compared to GlideSP alone. This suggests that ExptGMS is complementary to GlideSP. This is the same conclusion that can be drawn from Figure 2.

The reviewer also mentioned whether ExptGMS performs significantly better with certain datasets compared to others. Based on the above analysis, it can be observed that ExptGMS alone performs better for kinase and other enzymes than for protease. However, when considering GlideSP+ExptGMS, its performance for protease (top 10) is considered best over other targets.

In addition, to further facilitate the readers to explore Figure 2 in different perspectives, we have provided an additional supporting file containing the enrichment information for each target in the revised manuscript.

Fig2_Enrichment_by_target.csv (255 lines)

Target	PDBID	Target Type	TopN	ExptGMS	GlideSP	GlideSP+ExptGMS	alpha sphere (pocket shape)	MM/GBSA	USRCAT (3D similarity)	ECFP (2D similarity)	ExptGMS shape-only	TF3P (3D force field-based similarity)
AA2AR	3EML	GPCR	10	10	4	8	0	1	4	10	6	10
ABL1	2HZI	Kinase	10	8	9	9	2	1	1	8	3	8
ACE	3BKL	Protease	10	9	9	9	7	2	6	10	9	5
ACES	1E66	Other Enzym	10	2	3	0	0	3	5	8	0	6
ADA	2E1W	Other Enzym	10	7	1	5	0	0	10	10	3	10
ADRB1	2VT4	GPCR	10	10	6	8	0	2	2	10	3	10
ADRB2	3NY8	GPCR	10	7	7	8	0	0	4	8	5	10
AKT1	3CQW	Kinase	10	2	6	7	1	0	4	10	1	0
AKT2	3D0E	Kinase	10	10	6	7	0	5	5	10	7	0
ALDR	2HVS	Other Enzym	10	9	10	10	4	1	0	10	7	1
AMPC	1L2S	Other Enzym	10	2	1	0	0	0	1	10	0	0
ANDR	2AM9	Nuclear Rece	10	1	8	8	3	0	10	10	1	8

In summary, as mentioned in the manuscript, the pilot testing conducted on ExptGMS with a resolution of 3.0 Å confirmed our hypothesis that ExptGMS contains signals that are useful for improving active compound enrichment. To further maximize the effectiveness of such signals, we need to consider ExptGMS with multiple resolutions.

- In figure 2 the authors benchmark ExptGMS against several VS software programs that are either ligand or receptor based. They note in figure 3 that their method can improve false negative/positive hit detection. What was the overall false positive/negative rates between the methods used? Do the authors see a relationship between the number of top 10, 50, and 100 compounds and the false negative/positive rate? Does this depend on the choice of software (ex ligand vs receptor based VS)? It would be helpful if the authors could further elaborate on the false positive and negative hit detection rates between their method and the existing VS methods.

Response: We appreciate the reviewer's comments.

To address the reviewer's concern, we tested the overall false positive rate (FPR) and false negative rate (FNR) of the methods involved in this study (shown in below table), using the same configuration as figure 2 (i.e. ExptGMS was only used at a single resolution 3.0 Å).

Table A: FNR and FPR test on 85 DUD-E targets

Method	FNR*	FPR*
ECP (2D similarity)	0.68	0.02
GlideSP	0.74	0.02
GlideSP+ExptGMS	0.78	0.02
ExptGMS	0.82	0.02
ExptGMS shape-only	0.85	0.03
TF3P (3D force field-based similarity)	0.86	0.03
USRCAT (3D similarity)	0.89	0.03
MM/GBSA	0.90	0.03
alpha sphere (pocket shape)	0.92	0.03

Note: *Top N ranked molecules (where N equals to the number of actual positive samples for a target) of a method were considered as predicted positive samples for this method during the FNR and FPR calculation. The value is the average of 85 targets.

Based on the results displayed in above table, ExptGMS does not show lower FNR or FPR than GlideSP. However, when we test FNR and FPR for GBDT models (as shown in the table below), we can observe that "GlideSP and multi-resolution ExptGMS" exhibit a comparatively lower FNR than GlideSP.

Table B: FNR and FPR test on 12 DUD-E targets (testing set)

GBDT models with different features	FNR*	FPR*
GlideSP and multi-resolution ExptGMS	0.78	0.02
GlideSP and Alpha sphere	0.83	0.02
GlideSP and MM/GBSA	0.80	0.02
GlideSP and USRCAT	0.81	0.02
GlideSP and TF3P	0.82	0.02
GlideSP	0.80	0.02

Note: *Top N ranked molecules (where N equals to the number of actual positive samples for a target) of a method were considered as estimated positive samples for this method during the FNR and FPR calculation. The value is the average value of 12 targets.

How can the above two tables be understood? As mentioned in the manuscript, tests in Figure 2 using single resolution ExptGMS at 3.0 Å only allow us to observe weak signals. ExptGMS and GlideSP show comparable performance while using totally different principles for scoring. After confirming in Figure 5 that multi-resolution ExptGMS can complement each other, we built a GBDT model leveraging multi-resolution ExptGMS and observed relatively strong signals. This is the same situation for Table A and Table B. We observe weak signals in Table A for the single resolution

ExptGMS, while relatively strong signals are seen in Table B for the multi-resolution ExptGMS.

Regarding the reviewer's comments "do the authors see a relationship between the number of top 10, 50, and 100 compounds and the false negative/positive rate?", it is not appropriate to calculate top 10, 50 or 100 compounds for FNR or FPR. The reasons are listed below. The actual number of positive and negative samples for a target in DUD-E is approximately 400 and 12,000, respectively. This results in a very small false negative rate (FNR) and false positive rate (FPR) when using top N as predicted positive, especially when N is small. This is particularly true for N values of 10, 50, or 100. Furthermore, the positive-to-negative ratio of compounds in DUD-E is approximately 1:30. This means that even if we choose the top N compounds, where N is equal to the number of actual positive samples for a target, we will still have a very small false positive rate (FPR) for all the methods.

We use Figure 3 as case study to demonstrate why ExptGMS is potentially capable of detecting false positive and false negative molecules for GlideSP. As mentioned in the manuscript, Figure 2 and Figure 3 together only demonstrate that single resolution ExptGMS show weak signal that it may complement GlideSP. However, if we need to see changes in indecies, we need to find a way to strengthen such signal, i.e. the multi-resolution ExptGMS combined with GBDT model.

Given the limitation of FNR and FPR caused by the label imbalance issue and the consistency of conclusions drawn from analysing above FNR/FNR and our current method (Figure 2, 3, 5, and Table 1), it seems better to keep enrichment as index in the manuscript, which is also a commonly used and intuitive index for HTVS method testing.

3. We feel the 3CLpro in vitro assay description is unclear as it is written. Is this a peptide displacement assay? Further, what was the construct for the peptide/choices of fluorophores? Further explanation of how the in vitro assay was constructed and performed would be helpful.

Response: We appreciate the reviewer's comments.

It is not a peptide displacement assay but a peptide degradation inhibition assay. To improve clarity on the assasy discription, we have added below paragraph in the revised manuscript (lines 548 – 555).

SARS-COV-2 3CLpro (EC: 3.4.22.69) is a 3C-like proteinase that recognizes substrates

containing the core sequence [ILMVF]-Q-↓-[SGACN]^{3,4}. The inhibition potency of a potential inhibitor was determined by FRET-based assay using a FRET-compatible peptide substrate MCA-AVLQ↓SGFR-Lys (Dnp)-Lys-NH₂ (“↓” indicates the cleavage site). MCA fluorescence is initially quenched by the Dnp group until cleavage (at cleavage site) separates them. The maximum excitation light of MCA is 320 nm, while the maximum emission wavelength is 405 nm. The activity of 3CLpro was detected by measuring fluorescence.

4. In Figure 5, the authors describe one possibility for why ExptGMS with different resolutions complemented each other by using solvent exposure in the pocket. How are t-test results in box plots? We feel that it is difficult to judge whether the red and blue boxes in each resolution have statistically significant differences.

Comment 4 and 5 discuss a same issue: what is the principal reason that multi-resolution grids complemented each other. Therefore, we would like to respond them together (please check the response for comment #5).

5. In Table 1, the authors try to demonstrate the usefulness of multi-resolution analysis using a machine learning model. The table seems to show the advantage of the combination between GlideSP and multi-resolution ExptGSM. However, we are concerned about multiple hypothesis testing. This is also related to Figure 5. The authors tested more and more things about multi-resolution analysis but did not show a principled test. It would be helpful to get a deeper understanding about multi-resolution analysis if the authors could provide their thoughts and tests related to the principle.

Response: We thank the reviewer for raising this point.

Regarding exploring the principal reason that multi-resolution grids complemented each other, we performed the following two actions in the revised manuscript:

- *First, we did the t-test and found no statistically significant differences, which indicates that the variance of solvent exposure in the pocket may not be a reason to explain why ExptGMS with different resolutions complemented each other. Therefore, we deleted the related content (figure 5b, 5c).*
- *Second, we provide a paragraph and new figure 5b to 5e in the revised manuscript (lines 243 to 264) to describe our thoughts and provide case study for the readers to gain deep understanding about how multi-resolution ExptGMS works.*

“...The question arises as to why ExptGMS with different resolutions can complement each other in terms of enriching active compounds? One possible explanation is that ExptGMS with different resolutions intend to score ligands from different perspectives. Low-resolution grids focusing on scaffold-level information, whereas high-resolution grids focusing on R group of atomic level information. This distinction arises due to the intrinsic characteristic of X-ray or electron

diffraction-based density, where decreasing the resolution results in a more uniform intensity distribution with fewer details expressed in the density map. To illustrate this point, we present a case involving PDB ID 2HV5. Here, an active compound exhibits a similar binding mode and scaffold to the co-crystallized ligand of the protein (Fig. 5b). This active ligand (yellow) can be ranked in top 100 by using ExptGMS with 3.5 Å but not with 2.5 Å. To highlight the difference of ExptGMS grids at these two resolutions, we selected grids with strong intensities (i.e., over 3σ) and showed them side by side in Figure 5c. The 2.5 Å grid appears more fragmented, containing numerous blobs with high intensity (red grid points) than 3.5 Å grid. When scoring the original co-crystallized ligand (cyan), the fragmented blobs in the 2.5 Å grid exhibit a higher degree of matching with the ligand than the 3.5 Å grid (Fig. 5d). However, when scoring the active compound sharing similar scaffold but with different substitution groups, the 3.5 Å grid shows better match than the 2.5 Å grid. Figure 5e illustrates that the penalty introduced by the fragmented blob (#1) in the 2.5 Å grid is waived in the 3.5 Å grid, and the strong blob (#2) in the 2.5 Å grid spreads across a wider region, fitting more accurately with the scaffold profile of the compound.

Figure 5. Performance of electron density-based grid matching score (ExptGMS) with varying resolutions on 85 targets from the Directory of Useful Decoys–Enhanced (DUD-E) data set. (a) Performance comparison of ExptGMS at different resolutions. A target was labeled with a resolution-specific color if ExptGMS+GlideSP demonstrates more active compounds than GlideSP score alone, among any of the top 10, 50, or 100 ranked compounds for that target, at that particular resolution. (b) The binding mode of co-crystallized ligand and docked active compound (CHEMBL344526_63) in the pocket of PDB ID 2HV5. The co-crystallized ligand,

active compound, and pocket atoms are colored with cyan, yellow, and green, respectively. (c) ExptGMS grids of 2HV5 pocket at 2.5 Å and 3.5 Å. Only the grid points with ED intensity over 3.0 are shown. ExptGMS grids are colored with a rainbow scheme ranging from low (blue) to high ED intensity (red). The outlying blob (#1) and strong-grid-points-concentrated blob (#2) are indicated with green arrows to show the fragmentation and disparate distribution of 2.5 Å grids, respectively. (d) The match of co-crystallized ligand with ExptGMS grids at 2.5 Å and 3.5 Å. The molecule in cyan matches with 2.5 Å grid better than 3.5 Å. (e) The match of active compound (CHEMBL344526_63) with ExptGMS grids at 2.5 Å and 3.5 Å. The yellow molecule exhibits a better match with the 3.5 Å grid: it attains a top 100 ranking in ExptGMS at 3.5 Å, whereas it does not achieve a similar ranking at 2.5 Å. The penalty associated with the presence of blob #1 in the 2.5 Å grid is eliminated in the 3.5 Å grid. Additionally, the expansion of blob #2 in the 2.5 Å grid across a larger area in 3.5 Å grid makes 3.5 Å grid align more favorably with the compound's scaffold.

Hopefully, with the newly added Figure 5b to 5e as a case study, we can provide an explanation for why multi-resolution grids complement each other. Decreasing the resolution results in a more uniform distribution of grid values, which suggests a higher degree of tolerance for conformational matches with ligand candidates. Such characteristics affect the recall of compounds that differ significantly from the reference ligand topology and may, consequently, improve enrichment.

To address the reviewer's concern on multiple hypothesis testing, I explain our logic as below:

1. *Instead of testing GBDT and multi-resolution together, we in fact tested them one by one. We first confirmed that "ExptGMS with different resolutions can complement each other in terms of enriching active compounds" by using Figure 4, 5a, and 5b, and then raised the question: how to leverage the complementarity between ExptGMS with different resolutions. To answer this question, we introduced GBDT.*

2. *When demonstrating the superiority of multi-resolution ExptGMS over other methods, we used GBDT to support all the methods involved in the tests, not only for multi-resolution ExptGMS.*

6.

Minor points:

Supplementary materials are mentioned but not available under biorxiv posting.

Reviewed by CJ San Felipe, Hiroki Yamamura, and James Fraser (UCSF)

Response: We thank the reviewer for raising this point.

The supplementary materials will be added to biorxiv after the paper is published.

References

1. Pedregosa F, *et al.* Scikit-learn: Machine Learning in Python. *Journal of Machine Learning Research* **12**, 2825-2830 (2011).
2. Mysinger MM, Carchia M, Irwin JJ, Shoichet BK. Directory of useful decoys, enhanced (DUD-E): better ligands and decoys for better benchmarking. *J Med Chem* **55**, 6582-6594 (2012).
3. Jin Z, *et al.* Structure of M(pro) from SARS-CoV-2 and discovery of its inhibitors. *Nature* **582**, 289-293 (2020).
4. Zhang L, *et al.* Crystal structure of SARS-CoV-2 main protease provides a basis for design of improved alpha-ketoamide inhibitors. *Science* **368**, 409-412 (2020).

REVIEWERS' COMMENTS:

Reviewer #1 (Remarks to the Author):

I think that reviewer comments were generally well addressed by the authors, with the exception of analyzing false positive and negative rates raised by reviewer 2. In the rebuttal (but not in the revised manuscript) the authors shared false negative and false positive rates using a single score cutoff. A better approach is to plot the receiver operator characteristic (ROC) curve. This is a standard in the field, and it considers all possible score cutoffs. The area under the ROC curve is also a standard and very informative metric. I think such analysis would strengthen the statistics. Also, sharing the raw data from which these and other statistics can be calculated (scores and active/decoy labels for each system) is a good idea, making it possible for interested readers to take a closer look at the data.

Now that confidence intervals for enrichment factors of multi-resolution ExpGMS + Glide are reported, as well as the authors commenting on the lack of overfitting, the case for the superior performance of Glide + ExpGMS is more solid. The comment from reviewer 2 about true/false positive rates (and ROC curves, and the area under the ROC curves) still stands though.

Some minor notes:

Cross validation involves multiple rounds of validation, for multiple splits of the available data into multiple training/test sets. From the author's description what was done is just "validation", not "cross validation".

The authors write in revised SI:

"This involved using a validation set, which consisted of 10% of the training set data"
I hope the validation set was not part of the training set.

A description of how low resolution maps are calculated from higher affinity maps is missing.

Reviewer 2 wrote:

"the authors also included low pass filtered (maps calculated only using low resolution reflections)"

But I can't find this information in the manuscript.

Diogo Santos-Martins

Reviewer #2 (Remarks to the Author):

We have no further comments.

Reviewer #1 (Remarks to the Author):

I think that reviewer comments were generally well addressed by the authors, with the exception of analyzing false positive and negative rates raised by reviewer 2. In the rebuttal (but not in the revised manuscript) the authors shared false negative and false positive rates using a single score cutoff. A better approach is to plot the receiver operator characteristic (ROC) curve. This is a standard in the field, and it considers all possible score cutoffs. The area under the ROC curve is also a standard and very informative metric. I think such analysis would strengthen the statistics. Also, sharing the raw data from which these and other statistics can be calculated (scores and active/decoy labels for each system) is a good idea, making it possible for interested readers to take a closer look at the data.

Response: We appreciate the reviewer's comments.

First, we provided an additional file containing ExptGMS scores at different resolution, GBDT predicted probability, and active/decoy labels.

Supplementary Data3_Table1_raw_data.csv (15,424 lines)

target	name	smiles	label	ED25	ED30	ED35	ED45	ED55	GlideS	GlideSP	GlideSP	GlideSP	GlideSP	GlideSP	GlideSP	MM/GBSA (GBDT probability)
AMPC	109.49	O=C(O)C1CCCC1S(=O)(=O)Nc1ccc(Cl)cc1	active	15534.2525	16001.528	16308.812	18637.336	19876.649	-4.8	0.508	0.467	0.618	0.481	0.352	0.364	
AMPC	212.4	CSCC(C@H)C(O)O)N1C(=O)C2CC(C(=O)O)C	active	14712.72591	15019.927	15502.695	17668.519	19685.689	-6.6	0.532	0.532	0.492	0.688	0.558	0.397	
AMPC	317.12	Nc1onc(-c2ccc(Cl)cc2)c1C(=O)O)O	active	14864.3475	15215.478	16003.555	18194.943	19978.154	-6	0.468	0.479	0.628	0.704	0.431	0.373	
AMPC	104.58	O=C(O)C1CCCC1S(=O)(=O)Nc1ccc(Cl)cc1	active	14262.27833	14757.975	15826.28	17872.798	19554.08	-5.4	0.404	0.457	0.67	0.931	0.429	0.364	
AMPC	319.2	Cc1nc2ccc(P(=O)(O)O)cc2s1	active	16316.3	16827.899	17452.811	18756.396	20695.863	-5.8	0.497	0.457	0.619	0.393	0.51	0.387	
AMPC	203.32	O=C(O)C1CCCC2c(Cl)C(=O)N(C@H)C1ccc(O)cc1	active	16265.46462	16824.956	17078.234	18848.729	20503.073	-6.6	0.532	0.52	0.547	0.442	0.558	0.431	
AMPC	116.17	O=C(O)C1CCCC1NS(=O)(=O)C2=CC=CC=C2C(=O)O)O	active	15262.77409	15724.568	16176.943	17644.553	19619.122	-5.1	0.467	0.39	0.554	0.509	0.484	0.374	
AMPC	320.28	O=C(O)C1C(C@H)C(C@H)C(C@H)C1c1n[nH]c1=O	active	15900.192	15863.233	16369.224	18256.473	20000.261	-6	0.449	0.479	0.476	0.457	0.464	0.383	
AMPC	304.43	C(C@H)O)C(C@H)C(=O)O)N1C(=O)C2CCCC2C	active	15588.88222	16157.87	16827.66	18763.323	20607.051	-6.3	0.498	0.603	0.57	0.478	0.441	0.333	

Second, we have added AUROC metric to Table 1 of the revised manuscript. In addition, we also added the following paragraph (line 249 to 258):

"...In addition to enhancing the enrichment of active compounds within the top N ranked molecules, we sought to assess the impact of ExptGMS on the classification of active and decoy compounds. For evaluation, we utilized the area under the receiver operating characteristic curve (AUROC). The GBDT model incorporating both GlideSP score and ExptGMS features, demonstrated a higher AUROC compared to the model that solely utilized GlideSP score as a feature (Table 1), reflecting the classifier's improvement with the inclusion of ExptGMS. Nonetheless, it is important to acknowledge that this improvement is mild and the absolute AUROC value remains relatively low, indicating the need to incorporate ExptGMS features into more sophisticated models in future research..."

Table 1. Performance of GBDT models trained using different features on the DUD-E test set (N=12).

Features used in the model	Average number of Active Compound in Top N, with 90% Confidence Interval			Diversity ^{*b}	AUROC ^{*d}
	N = 10	N = 50	N = 100		
GlideSP and multi-resolution ExptGMS ^{*a}	5.4 [5.0, 5.8]	21.6 [20.4, 22.9]	33.2 [30.9, 35.1]	0.64 [0.60, 0.67]	0.66

GlideSP and TF3P	5.4 [4.9, 5.8]	19.5 [18.1, 20.8]	27.7 [26.0, 29.6]	0.62 [0.60, 0.64]	0.64
GlideSP and MM/GBSA	4.5 [4.0, 5.2]	18.5 [17.0, 19.9]	30.1 [28.1, 32.2]	0.66 [0.62, 0.68]	0.65
GlideSP and USRCAT	4.5 [3.9, 5.1]	18.1 [16.9, 19.3]	30.4 [28.0, 32.3]	0.63 [0.59, 0.66]	0.63
GlideSP and Alpha sphere	5.1 [4.6, 5.6]	18.4 [17.0, 19.8]	29.2 [27.2, 31.0]	0.64 [0.61, 0.68]	0.63
GlideSP ^{*c}	4.3 [3.8, 4.8]	17.3 [16.1, 18.6]	29.9 [27.9, 31.9]	0.66 [0.62, 0.68]	0.62

**a:* Multiresolution indicates ExptGMS at resolutions of 2.5, 3.0, 3.5, 4.5, and 5.5 Å.

**b:* Diversity of active compounds among the top 100 ranked compounds.

**c:* Ranking was performed directly using the GlideSP score without using a trained model.

**d:* AUROC was calculated by using GBDT predicted probability of being an active compound.

Now that confidence intervals for enrichment factors of multi-resolution ExpGMS + Glide are reported, as well as the authors commenting on the lack of overfitting, the case for the superior performance of Glide + ExpGMS is more solid. The comment from reviewer 2 about true/false positive rates (and ROC curves, and the area under the ROC curves) still stands though.

Response: These concerns are the same with those mentioned in the previous question. Please check the response listed above.

Some minor notes:

Q1. Cross validation involves multiple rounds of validation, for multiple splits of the available data into multiple training/test sets. From the author's description what was done is just "validation", not "cross validation".

Response: We appreciate the reviewer's comments.

We have updated the wording in the legend of Figure S4 accordingly, with details shown in the response of Q2.

Q2. The authors write in revised SI:

"This involved using a validation set, which consisted of 10% of the training set data"

I hope the validation set was not part of the training set.

Response: We appreciate the reviewer's comments.

The validation set was not used during training.

We have updated the legend of Figure S4 in Supporting Information as below to make this point clear:

“...Figure S4. Training curve of GBDT (GlideSP and multi-resolution ExptGMS). The performance of GBDT on the training set is shown through a plot showing the loss versus iterations. To prevent overfitting, a validation set comprising 10% of the training data was created. It is important to note that the data from the validation set were not used during the training process, but rather reserved for the purpose of early stopping to mitigate overfitting. The training was initially set to iterate 500 steps. As shown in the above figure, it stopped at step 63. During the training process, it is observed that the validation loss initially decreases and subsequently stabilizes, without showing signs of an upward trend. This pattern suggests the absence of overfitting in the model...”

Q3. A description of how low resolution maps are calculated from higher affinity maps is missing. Reviewer 2 wrote:

"the authors also included low pass filtered (maps calculated only using low resolution reflections)"

But I can't find this information in the manuscript.

Response: We appreciate the reviewer's comments.

In the original manuscript, we briefly described this information in “Experimental ED Map Preparation and ExptGMS Grid Generation” part of the Method section. To further address the reviewer's concern, we added below details in the revised version (line 359-362):

“...Specifically, the highest available resolution X-ray diffraction data in .mtz format were used as input for phenix.fft to generate electron density maps at various resolutions, which were specified by the parameter d_min....”

Reviewer #2 (Remarks to the Author):

We have no further comments.